# Curvature Enhanced Manifold Sampling

## Abstract

Over-parameterized deep learning models, characterized by their large number of parameters, have demonstrated remarkable performance in various tasks. Despite the potential risk of overfitting, these models often generalize well to unseen data due to effective regularization techniques, with data augmentation being one of the most prominent methods. This strategy has proven effective in classification tasks, where label-preserving transformations are applicable. However, the application of data augmentation in regression problems remains underexplored. Recently, a new *manifold learning* approach for sampling synthetic data has been introduced, and it can be viewed as utilizing a first-order approximation of the data manifold. In this work, we propose to extend this direction by providing the fundamental theory and practical tools for approximating and sampling general data manifolds. Further, we introduce the curvature enhanced manifold sampling (CEMS) data augmentation method for regression. CEMS is based on a second-order encoding of the manifold, facilitating sampling and reconstruction of new points. Through extensive evaluations on multiple datasets and in comparison to several state-of-the-art approaches, we demonstrate that CEMS is superior in in-distribution and out-of-distribution tasks, while incurring only a mild computational overhead.

## 1 Introduction

Deep neural networks have demonstrated remarkable performance across a wide range of applications in various fields (Krizhevsky et al., 2012; Long et al., 2015; Mnih et al., 2015; Noh et al., 2015; Vinyals et al., 2015; He et al., 2016; Nam & Han, 2016; Wu et al., 2016). Despite their success, these models are often significantly over-parameterized, meaning they possess more parameters than the number of training examples. As a result, deep neural networks are prone to overfitting, whereby they "memorize" the training set rather than learning generalizable patterns, thus compromising their performance on unseen data. Regularization techniques are crucial to address this issue, as they modify the learning process to prevent overfitting by reducing the variance and increasing the bias of the underlying model (Goodfellow, 2016). Classical regularization methods, such as weight decay, dropout (Srivastava et al., 2014), and normalization techniques like Batch Normalization (Ioffe & Szegedy, 2015) and Layer Normalization (Ba et al., 2016), have been effective in many scenarios. In addition to these methods, recent research has explored the potential of data augmentation (DA) as a form of regularization. In this paper, we focus on the problem of regularizing *regression models* via *data augmentation*. That is, we explore how to artificially expand the train set (DA) for models that predict continuous values (regressors) to improve generalization and robustness.

Early work in modern computer vision revealed the effectiveness of basic image transformations such as translation and rotation (Krizhevsky et al., 2012), promoting data augmentation to become one of the key components in designing generalizable learning models (Shorten & Khoshgoftaar, 2019). In particular, classification tasks, whose goal is to predict a discrete label, benefited notably from the rapid development of DA (Simonyan & Zisserman, 2015; DeVries, 2017; Zhang et al., 2018; Zhong et al., 2020). The discrete and categorical nature of classification labels makes it easier to define label-preserving transformations and apply interpolations without compromising data integrity. In contrast, regression tasks, where the outputs are continuous, face unique challenges in ensuring that transformations produce valid input-output pairs and that interpolations maintain the underlying functional relationships. While certain regression challenges have adopted standard data augmentation approaches successfully (Redmon et al., 2016), existing DA methods are generally less effective for regression problems (Yao et al., 2022). For this reason, developing data augmentation tools for general regression problems is an emerging field of interest with a relatively small

number of available effective techniques. One of the recent state-of-the-art (SOTA) works introduced FOMA, a data-driven and domain-independent approach based on the theory and practice of manifold learning (Kaufman & Azencot, 2024). Our work is also inspired by manifold learning, where we consider DA as a *manifold approximation and sampling* challenge.

Manifold learning is fundamental to modern machine learning primarily through the *manifold hypothesis* (Belkin & Niyogi, 2003; Goodfellow, 2016), where complex and high-dimensional data is assumed to lie close or on an associated low-dimensional manifold. Multiple works leveraged the relation between data and manifolds (Zhu et al., 2018; Ansuini et al., 2019), and particularly, FOMA (Kaufman & Azencot, 2024) can be viewed as a method for generating new examples by sampling from the tangent space of the data manifold, approximated using the training distribution. The tangent space at a point is a linear approximation of the manifold at that point (Lee, 2012), and thus, FOMA is a first-order approach. However, while first-order approximations work well for relatively simple or well-behaved data, they often fall short when dealing with complex, curved real-world data. We demonstrate this effect in Fig. 1B-C, where first-order approximations of points with high curvature fail to capture the structure of the manifold. While it is natural to consider higher-order approximations for improving FOMA, their computational burden may be too limiting. In this work, we advocate that *second-order* manifold representations offer a compelling trade-off between effectiveness and compute requirements for data augmentation for regression problems.

We propose the curvature enhanced manifold sampling (CEMS) approach, which generates new examples by drawing from a second-order representation of the data manifold. Particularly, CEMS randomly generates points in the tangent space of the manifold, whereas FOMA is different as it scales down the orthogonal complement of the tangent space which captures how the manifold deviates from the linear approximation. FOMA and CEMS are data-driven and domain-independent, i.e., their samples are based on the underlying data distribution, whose domain can be arbitrary, e.g., time series, tabular, images, and other data forms. The inclusion of curvature information allows CEMS to better capture the intrinsic geometry of the data manifold, as it accounts for the non-linearities and complex structures that first-order methods might miss. In general, second-order methods are infeasible in modern machine learning due to computational costs incurred by high-dimensional vectors. Nevertheless, our analysis shows that CEMS is governed by the *intrinsic dimension* $d$ of the manifold, and its value is much smaller than the data dimension $D$, i.e., $d \ll D$. We extensively evaluate CEMS and show it is competitive in comparison to SOTA data augmentation approaches on in-distribution and out-of-distribution tasks. The contributions of our work can be summarized as follows:

1. We consider data augmentation for regression as a manifold learning problem, extending and generalizing prior approaches through providing the foundational theory and practice.

2. We introduce CEMS, a novel fully-differentiable, data-driven and domain-independent data augmentation technique that is based on a second-order approximation of the data manifold.

3. Across nine datasets, featuring numerous large-scale, real-world in-distribution and out-of-distribution tasks, we demonstrate that CEMS performs competitively or even surpasses other augmentation strategies.

## 2 RELATED WORK

The theoretical foundation for data augmentation (DA) is related to the study of Empirical Risk Minimization (ERM) (Vapnik, 1991) vs. Vicinal Risk Minimization (VRM) (Chapelle et al., 2000). In VRM, one considers an extended distribution to train on, in comparison to ERM, where the train distribution is used. Data augmentation is a common approach for extending data distributions through creating artificial samples. With the increased dependence of deep models on large volumes of data, DA has become a cornerstone in enhancing the performance and generalization of neural networks (DeVries, 2017; Chen et al., 2020b; Feng et al., 2021; Yang et al., 2022). Early work focused on domain-dependent augmentations for image, audio, and natural language data (Krizhevsky et al., 2012; He et al., 2016; Huang et al., 2017; Kobayashi, 2018; Park et al., 2019; Zhong et al., 2020). Later, automatic augmentation tools have been proposed (Cubuk et al., 2019; Lim et al., 2019), including domain-dependent search spaces for transformations. Still, adapting these methods to new

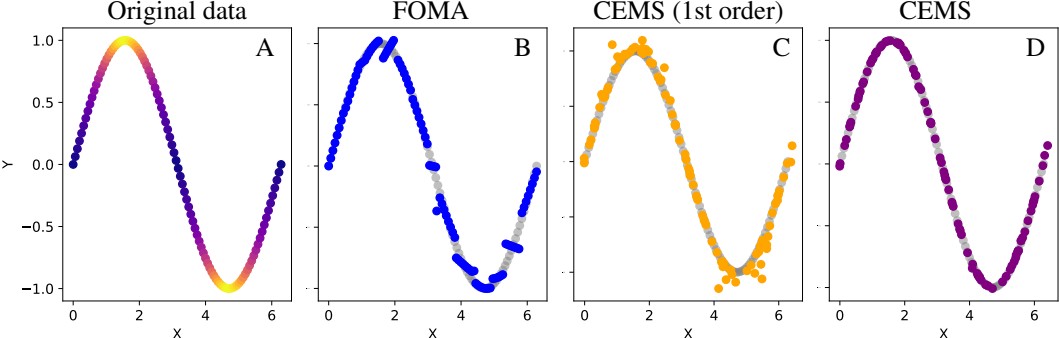

Figure 1: We demonstrate the effect of sampling from a one-dimensional manifold embedded in a two-dimensional space. A) The original data representing a sine wave where the color of each point represents the curvature at that point (brighter means higher curvature). B) Sampling using FOMA. C) Sampling using a first-order approximation. D) Sampling using CEMS (our approach).

data domains remains a challenge. This has sparked interest in developing domain-independent approaches that make minimal assumptions about the data domain, and it is the focus our research.

**DA for Classification.** Zhang et al. (2018) introduced mixup, a well-known domain-independent DA method that convexly combines pairs of input samples and their one-hot label representations during training. Following their work, a plethora of mixup-based techniques have been suggested such as manifold mixup (Verma et al., 2019) which extends the idea of mixing examples to the latent space. CutMix (Yun et al., 2019) implants a random rectangular region of the input into another and many others (Guo et al., 2019; Hendrycks et al., 2020; Berthelot et al., 2019; Greenewald et al., 2023; Lim et al., 2022). Recently, Erichson et al. (2024) extended the work (Lim et al., 2022) via stable training and further noise injections. While the family of mixup techniques have been shown to consistently improve classification learning systems (Cao et al., 2022), its efficacy is inconsistent on regression tasks (Yao et al., 2022).

**DA for Regression.** Unfortunately, there has been considerably less focus on developing data augmentation methods for regression tasks in comparison to the classification setting. Due to the simplicity and effectiveness of mixup-based tools in classification, a growing body of literature is drawn to adapting and extending the mixing process for regression. For instance, RegMix (Hwang & Whang, 2021) learns the optimal number of nearest neighbors to mix per sample. C-mixup (Yao et al., 2022) employs a Gaussian kernel to create a sampling probability distribution for each sample, taking label distances into account, and selecting samples for mixing according to this distribution. Anchor Data Augmentation (Schneider et al., 2023) clusters data points and adjusts the original points either towards or away from the cluster centroids. R-Mixup (Kan et al., 2023) focuses on enhancing model performance specifically for biological networks, whereas RC-Mixup (Hwang et al., 2024) extends C-mixup to be more robust against noise. Perhaps closest to our work is the recent FOMA method (Kaufman & Azencot, 2024) that does not rely on mixing samples, but rather, it samples from a first-order approximation of the data manifold. Still, to the best of our knowledge, our work is first in suggesting fundamental manifold learning theory and tools for DA, accompanied by an effective second-order augmentation technique.

**Manifold Learing.** Manifold learning has been a fundamental research area in machine learning, aiming to discover the intrinsic low-dimensional structure of high-dimensional data. While early work focused on dimensionality reduction of points and preserving their geometric features (Tenenbaum et al., 2000; Roweis & Saul, 2000; Belkin & Niyogi, 2003; Weinberger & Saul, 2004; Zhang & Zha, 2005; Coifman & Lafon, 2006), modern approaches also considered regularization (Ma et al., 2018; Zhu et al., 2018), explainable artificial intelligence (Ansuini et al., 2019; Kaufman & Azencot, 2023), and autoencoding (Chen et al., 2020a), among other applications. Recent advancements in manifold learning have enhanced anomaly detection and out-of-distribution (OOD) recognition. (Li et al., 2024) leveraged submanifold geometry, estimating tangent spaces and curvatures to define in-distribution regions for OOD detection. Gao et al. (2022) proposed a hyperbolic feature augmentation method, using the Poincaré ball model for distribution estimation and infinite

sampling, improving few-shot learning performance. Humayun et al. (2022) proposed MaGNET, a framework that enables uniform sampling on data manifolds derived from generative adversarial networks providing a retraining-free solution for data augmentation. Similarly, Chadebec & Allassonnière (2021) introduced a geometry-aware variational autoencoder that leverages second-order Runge–Kutta schemes for effective data generation in low-sample-size scenarios. Extending these ideas, Cui et al. (2023) presented a trajectory-aware principal manifold framework for image generation and data augmentation, which aligns sampled data with learned projection indices to improve representation and synthesis quality.

## 3 BACKGROUND

### 3.1 MANIFOLD LEARNING

A manifold $\mathcal{M} \subset \mathbb{R}^d$ is a mathematical structure that locally resembles an Euclidean space near each of its points (Lee, 2012). A ubiquitous assumption in machine learning states that high-dimensional point clouds $Z \subset \mathbb{R}^D$ satisfy the manifold hypothesis. Namely, the data $Z$ lie on a manifold $\mathcal{M}$ whose intrinsic dimension $d$ is significantly lower than the extrinsic dimension of the ambient space $D$, i.e., $d \ll D$ (Goodfellow, 2016). Manifold learning is a field in machine learning that develops theory and tools for analyzing and processing high-dimensional data under the lens of geometric manifolds.

### 3.2 CURVATURE-AWARE MANIFOLD LEARNING

Our curvature enhanced manifold sampling (CEMS) data augmentation method is based on a second-order approximation of the data manifold. There are several existing practical approaches for parameterizing a manifold given a collection of data points $Z = \{z^1, z^2, \cdots, z^N\} \subset \mathbb{R}^D$. Here, we focus on the curvature-aware manifold learning (CAML) algorithm (Li, 2018), since it scales to high-dimensional problems, it is numerically stable, and it is easy to code. Below, we include the necessary details for describing our approach, and we refer the reader to (Lee, 2012; 2018; Li, 2018) for additional details on Riemannian geometry and its realization in machine learning.

Following our discussion above, we assume $Z$ satisfies the manifold hypothesis. Formally, it means that there exists an embedding map $f : \mathcal{M} \to \mathbb{R}^D$ such that

$$z^i = f(u^i), \quad i = 1, \ldots, N, \tag{1}$$

where $U = \{u^1, u^2, \cdots, u^N\} \subset \mathbb{R}^d$ are low-dimensional representations of $Z$. In practice, CAML parameterizes $f$ by projecting $z \in Z$ to its tangent and normal spaces at $u \in \mathcal{M}$, where the tangent space is obtained by a linear transformation and the normal space is provided via a second-order local approximation. Specifically, given a point $z \in Z$, we find close points $N_z = \{z_j\}_{j=1}^k$, forming the neighborhood of $z$. Next, we construct an orthonormal basis $B_u := [B_{\mathcal{T}_u}, B_{\mathcal{N}_u}]$ for the tangent space $\mathcal{T}_u\mathcal{M}$ and the normal space $\mathcal{N}_u\mathcal{M}$ at $u$. We then project $N_z$ and $z$ onto $B_{\mathcal{T}_u}$ and $B_{\mathcal{N}_u}$, yielding $U_z = \{u_j\}_{j=1}^k$, $u$ and $G_z = \{g_j\}_{j=1}^k$, $g$ respectively. To allow arbitrary sampling from $\mathcal{M}$, we assume that $g$ is a map from the tangent space to the normal space, i.e., $g : \mathcal{T}_u\mathcal{M} \to \mathcal{N}_u\mathcal{M}$. The second-order Taylor expansion of $g$ around a point $u \in \mathcal{M}$ is given by

$$g(u_j) = g(u) + (u_j - u)^T \nabla g(u) + \frac{1}{2}(u_j - u)^T H(u)(u_j - u) + \mathcal{O}(|u_j|_2^2), \tag{2}$$

where the linear (gradient) and quadratic (Hessian) terms are unknown. To compute $\nabla g(u)$ and $H(u)$, one needs to collect the linear coefficients and constants arising from Eq. 2 into matrices $\Psi$ and $G$, respectively, solve a linear system of equations (Eq. 9), and extract the numerical estimates of the gradient and Hessian. Finally, we can map the pair $(u, g)$ back into its original space $z \in Z$ by computing $z := f(u) = B_u[u, g(u)]$. See also App. A for additional details.

## 4 CURVATURE ENHANCED MANIFOLD SAMPLING

We assume to be given a regression training set $\mathcal{D} := \{(x^i, y^i)\}_{i=1}^N$, where $x^i$ is the data sample and $y^i$ is its corresponding prediction, and we denote by $z^i = [x^i, y^i] \in \mathbb{R}^D$ the concatenation

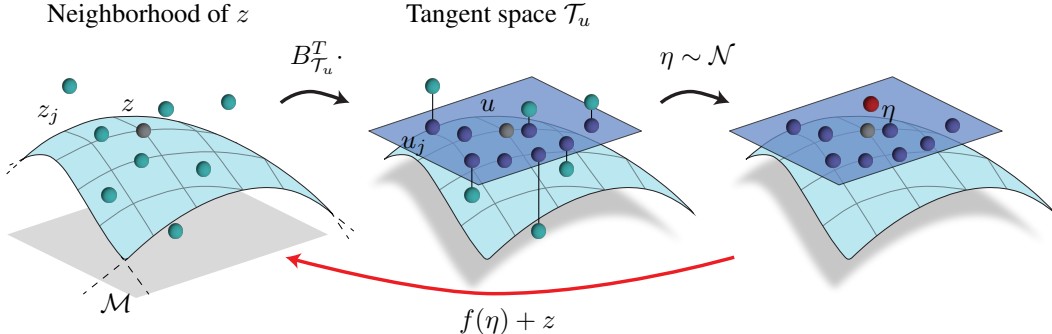

Figure 2: CEMS forms the neighborhood for every point $z$ (left), computes a basis $B_{\mathcal{T}_u}$ for the tangent space via SVD (middle) while obtaining an estimate for the embedding map $f$, samples a new point $\eta$ close to $u$ (right), and un-projects it back to $\mathbb{R}^D$ using $f$.

of $x^i, y^i$. During training, given a mini-batch $Z = \{z^{i1}, \cdots z^{ib}\}$, where $b$ is the batch size, we perform the following procedure for each $z \in Z$ to create a new sample $\tilde{z}$, omitting the superscript $i$ to simplify notation. To account for the discrepancies in scale between $X$ and $Y$, we normalize $Y$ to the range $[0, 1]$. Our curvature enhanced manifold sampling (CEMS) augmentation approach consists of four main steps: 1) Extract a neighborhood $N_z$ from $\mathcal{D}$ for every point $z$; 2) Construct a basis $B_u := [B_{\mathcal{T}_u}, B_{\mathcal{N}_u}]$ for the tangent space $\mathcal{T}_u\mathcal{M}$ and the normal space $\mathcal{N}_u\mathcal{M}$ and project the neighborhood onto it; 3) Form and solve the linear system of equations in Eq. 9 to obtain the parameterization $g$; 4) Sample a new point from $\mathcal{T}_u$, evaluate its $g$ via Eq. 2, and un-project it onto the ambient space using $f$. Below, we detail how we perform each step, and we motivate our design choices. Pseudo-code and illustration of CEMS are given in Alg. 1 and Fig. 2.

**Neighborhood extraction.** The approach we consider in this work is *local* in the sense that we represent the manifold structure in the vicinity of a specific point $z \in \mathcal{D}$ or within its neighborhood $N_z$. High-quality approximations of local properties of the manifold $\nabla g(u)$ and $H(u)$ depend directly on the proximity of the elements in $N_z$ to $z$. A straightforward approach to extracting $N_z$ is to compute the $k$-nearest neighbors (kNN) (Cover & Hart, 1967) of $z$. Specifically, we construct neighborhoods in the joint input-output space $\mathcal{X} \times \mathcal{Y}$ to align with the manifold hypothesis, preserving the local continuity of the data and avoiding the artificial separations introduced by clustering-based methods. For each point $z_i \in \mathcal{Z}$, $N_{z_i}$ is defined as the $k$ closest points in feature space, which ensures the local geometry is captured reliably while minimizing diversity within the neighborhoods. Furthermore, our reliance on the local-Euclidean prior assumes that the manifold is sufficiently smooth at small scales, justifying the validity of linear approximations such as those employed by our method. This assumption underpins the extraction of neighborhood sets and their utility in manifold analysis, as it guarantees that the neighborhoods respect the underlying manifold structure.

Our analysis below and in Sec. B shows that the computational complexity of CEMS is governed by singular value decomposition (SVD) calculations, required for basis construction. To reduce runtime, $\mathcal{D}$ can be pre-processed, storing $\nabla g(u)$ and $H(u)$ for every $z := f(u)$ on the disk, as these properties remain unchanged during training. While this pre-computation reduces runtime significantly, it incurs high memory complexity, $\mathcal{O}(2d(D - d))$, and the choice of neighbors is fixed during training. To address these limitations, we use the *same neighborhood* for all points in $N_z$, re-using neighborhoods and basis computations for every $z_j \in N_z$. This improves efficiency, though at the cost of accuracy, since every $z_j \in N_z$ is assumed to share the same neighborhood. Finally, the batch size determines the number of neighbors for each point, providing a balance between computational efficiency and accuracy.

**Basis construction and projection.** To find an orthonormal basis for the tangent space $\mathcal{T}_u\mathcal{M}$ and the normal space $\mathcal{N}_u\mathcal{M}$, we follow standard approaches (Singer & Wu, 2012; Li, 2018) that utilize the singular value decomposition (SVD). Specifically, we compute SVD on the centered points $\{z_j - z\}_{j=1}^k = USV^T$, while keeping our pipeline to be fully differentiable (Ionescu et al., 2015). Importantly, SVD is calculated once for every batch, as discussed above. The first $d$ columns of $U$ determine the basis for the tangent space, i.e., $B_{\mathcal{T}_u} := U[:, 1{:}d]$ and the last $D - d$ columns determine the basis for the normal space $B_{\mathcal{N}_u} := U[:, d + 1{:}D]$ such that $B_u := [B_{\mathcal{T}_u}, B_{\mathcal{N}_u}]$ is the concatenation of the bases. While the intrinsic dimension $d$ can be viewed as a hyper-parameter

of CEMS, we estimate it in practice using a robust estimator (Facco et al., 2017). The centered neighbors $z_j - z$ are projected to the tangent space and the normal space via $u_j := B_{\mathcal{T}_u}^T \cdot (z_j - z)$, $g(u_j) := B_{\mathcal{N}_u}^T \cdot (z_j - z)$, respectively, where $A^T$ is the transposed matrix of $A$, and $A \cdot v$ is a matrix-vector multiplication. Centering the points around $z$ map the point $u$ to the zero vector, and thus, Eq. 2 is transformed to the following approximation:

$$g(u_j) = u_j^T \nabla g(u) + \frac{1}{2} u_j^T H(u) u_j \ . \tag{3}$$

**Linear system of equations.** Under the change of basis $B_u$, we form the matrices $\Psi$ and $G$ as described in App. A, containing $\{u_j\}_{j=1}^k$ and $\{g(u_j)\}_{j=1}^k$, respectively. We then solve Eq. 9 via differentiable least squares, and we obtain an estimation of $\nabla g(u)$ and $H(u)$, allowing to map new points in the vicinity of $u$ by computing $g$. Note that while $N_z$ and $B_u$ are shared across the batch, the linear solve is still computed separately per point.

**Sampling and un-projecting.** To generate new examples using the above machinery, we need to sample a point $\eta \in \mathbb{R}^d$ from the neighborhood of $u$, and un-project it to the ambient space $\mathbb{R}^D$ (through the parameterization $g$ and map $f$). While various sophisticated sampling techniques could be devised, we opted for a simple sampler with a single hyperparameter. In practice, we draw $\eta \sim \mathcal{N}(0, \sigma I_d)$. To un-project $\eta$ back to the original space, we compute

$$z_\eta := f(\eta) = B_u \cdot [\eta, g(\eta)] + z \ . \tag{4}$$

**Adaptation to batches.** For completeness, we also describe briefly the adaptation of CEMS to the training setting where we utilize mini-batches. As mentioned above, given $z$ and its neighborhood $N_z$, we re-use the same neighborhood and subsequent basis computations for every $z_j \in N_z$. This adaptation requires a small modification to the method. 1) We include the point $z := z_0$ in the neighborhood $N_z = \{z_j\}_{j=0}^k$. 2) We find an orthonormal basis $B_u$ that spans $N_z - \mu$, where $\mu$ is the mean of $N_z$. 3) After projecting to the coordinates of $B_u$, we get $U_z = \{u_j\}_{j=0}^k$ and $G_z = \{g_j\}_{j=0}^k$. For every point $u_j$, we gather a set of close points and their embeddings via $g$. 4) In contrast to the point-wise basis estimation, where $u$ served as the origin (zero vector), in the batch-wise computation we need to account for $u_j$ and $g(u_j)$ in Eq. 2. While steps 5-7 in Alg. 1 remain unchanged, at step 8 we sample a point $\eta$ near $u_l$: $\eta \sim \mathcal{N}(u_l, \sigma I_d)$. Step 9 changes to $g(\eta_l) = g(u_l) + (\eta_l - u_l)^T \nabla g + \frac{1}{2}(\eta_l - u_l)^T H(\eta_l - u_l)$ and step 10 changes to $z_{\eta_l} := f(\eta_l) = B_u \cdot [\eta_l, g(\eta_l)] + \mu_{N_z}$. A full description of the algorithm appears in Alg. 2.

**Complexity analysis.** There are two computationally demanding calculations used by CEMS, SVD and least squares. Given a data mini-batch $Z \in \mathbb{R}^{b \times D}$, where $b$ is the batch size. Then, SVD requires $\mathcal{O}(\min(bD^2, Db^2))$ operations, whereas the solution of under-determined least squares costs $\mathcal{O}(b^2 d^2)$. Using the manifold hypothesis, we assume that $d \ll D$ therefore $d \in \mathcal{O}(D^2)$ and thus, the overall time complexity of CEMS is given by $\mathcal{O}(b^2 D)$ which is proportional to the mabient dimension $D$. See also App. B for a more detailed analysis.

---

**Algorithm 1** Curvature Enhanced Manifold Sampling (CEMS$_p$)

---

**Require:** Training data $Z = \{z^i = [x^i, y^i]\}_{i=1}^N$. A sample $z \in Z$
 1:    Find K-nearest neighbors $N_z$ of $z$
 2:    Find an orthonormal basis $B_u$ that spans $N_z - z$
 3:    Project every $z_j - z$ to the local orthonormal coordinates:
 4:       $u_j = B_{\mathcal{T}_u}^T \cdot (z_j - z)$, $g_j = B_{N_u}^T \cdot (z_j - z)$
 5:    Construct $G$ and $\Psi$ as in Eq. 9
 6:    Solve $\Psi A = G$
 7:    Extract $\nabla g(z)$ and $H(z)$ from $A$
 8:    Sample noise $\eta \sim \mathcal{N}(0, \sigma I_d)$
 9:    Calculate $g(\eta) = \eta^T \nabla g + \frac{1}{2} \eta^T H \eta$
10:    Un-project $\eta$ back to the original coordinates, $z_\eta := f(\eta) = B_u \cdot [\eta, g(\eta)] + z$
11: **return** $z_\eta$

---

**Memory analysis.**  The memory requirements of CEMS are primarily dictated by the computation of the SVD. Notably, the SVD is computed independently for each batch rather than for the entire dataset. In our PyTorch implementation, we leverage the economy/reduced SVD variant, which significantly reduces memory usage compared to the full SVD. For a batch matrix of size $b \times D$ (where $b$ is the batch size and $D$ is the ambient dimension), the space complexity is $O(bD + \min(b, D)(b + D))$. This is substantially more efficient than the full SVD, which requires $O(bD + b^2 + D^2)$ memory. In practice, CEMS is particularly effective in scenarios where the batch size $b$ is much smaller than the ambient dimension $D$ (common in deep learning), resulting in a memory complexity that is approximately proportional to $D$.

**Comparison with FOMA.**  FOMA (Kaufman & Azencot, 2024) can be interpreted as a special case of CEMS. Specifically, we can describe FOMA using our notations as follows: given a sample $z$ and its neighborhood $N_z$, FOMA constructs a basis $B_u := [B_{\mathcal{T}_u}, B_{\mathcal{N}_u}]$ for the tangent space $\mathcal{T}_u \mathcal{M}$ and the normal space $\mathcal{N}_u \mathcal{M}$ and projects the neighborhood onto it, yielding $U_z = \{u_j\}_{j=1}^k$ and $G_z = \{g_j\}_{j=1}^k$, respectively. Rather than estimating the gradient $\nabla g(u_j)$ and Hessian $H(u_j)$ at each point $u_j \in U_z$ and then sampling using the Taylor expansion as performed in CEMS, FOMA generates new samples by scaling down $G_z$. That is, for each $z_j \in N_z$, the corresponding $\tilde{g}_j = \lambda g_j$ is scaled where $\lambda \in (0, 1)$. To complete the sampling process, every $u_j$ is un-projected back to the original coordinates by computing $\tilde{z} := f(u_j) = B_u \cdot [u_j, \lambda g(u_j)]$. Therefore, FOMA does not use the embedding map $g$ as detailed in Eq. 2, but it samples random points instead. Unfortunately, this sampling technique may yield new points that are not on the data manifold, especially on highly curved locations, as is also illustrated in Fig. 1.

### 4.1 Approximation Error Bounds

In what follows, we provide a theoretical justification for the sampling error of CEMS in comparison to first-order approaches. Let $f : \mathbb{R}^d \to \mathbb{R}^D$ be a twice-differentiable function, we can express the Taylor expansion around a point $u_0$ up to first and second order as follows,

$$f^{(1)}(u) = f(u_0) + \nabla f(u_0)^T (u - u_0) \, , \tag{5}$$

$$f^{(2)}(u) = f(u_0) + \nabla f(u_0)^T (u - u_0) + \frac{1}{2}(u - u_0)^T H_f(u_0)(u - u_0) \, . \tag{6}$$

Under standard smoothness assumptions, the approximation errors can be bounded as follows:

**Theorem 4.1** (Error Bounds). *(Fowkes et al., 2013) For a twice-differentiable function $f$ with Lipschitz continuous Hessian in a neighborhood of $u_0$, we have that*

$$\|f(u) - f^{(1)}(u)\| \leq \frac{M}{2}\|u - u_0\|^2, \quad \|f(u) - f^{(2)}(u)\| \leq \frac{L}{6}\|u - u_0\|^3 \, , \tag{7}$$

*where $M$ bounds the spectral norm of $H_f(u)$ and $L$ is the Lipschitz constant of $H_f(u)$ in the neighborhood of $u_0$.*

The second-order error decreases as $O(\|u - u_0\|^3)$ compared to $O(\|u - u_0\|^2)$ for first-order methods. This faster convergence rate ensures more accurate sampling in the vicinity of training points.

## 5 Experiments

### 5.1 Sine example

Real-world data is often complex and curved, exhibiting intricate patterns that cannot be adequately captured by linear or simplistic models. By employing higher-order approximations of the manifold, we can generate samples that align with the true nature of real-world data. In Fig. 1, we demonstrate a toy sine example, highlighting the differences between first-order and second-order approaches. Specifically, we generated a two-dimensional point cloud of a sine wave whose intrinsic dimension is one (Fig. 1, left). Then, we sampled points from this distribution using mini-batches from the train set and various data augmentation techniques. The first-order method, FOMA (Kaufman & Azencot, 2024), struggles to adhere to the curvature of the manifold in highly-curved points, as can be seen

Table 1: Comparison of in-distribution generalization tasks. Values in bold indicate the best results, while underlined values represent the second best. We present the average RMSE and MAPE across three seeds. Detailed results, including standard deviation, are available in App H.

| | Airfoil | | NO2 | | Exchange-Rate | | Electricity | |
|---|---|---|---|---|---|---|---|---|
| | RMSE | MAPE | RMSE | MAPE | RMSE | MAPE | RMSE | MAPE |
| ERM | 2.901 | 1.753 | 0.537 | 13.615 | 0.024 | 2.423 | 0.058 | 13.861 |
| Mixup | 3.730 | 2.327 | 0.528 | 13.534 | 0.024 | 2.441 | 0.058 | 14.306 |
| Mani Mixup | 3.063 | 1.842 | 0.522 | 13.382 | 0.024 | 2.475 | 0.058 | 14.556 |
| C-Mixup | 2.717 | 1.610 | 0.509 | 12.998 | 0.020 | 2.041 | **0.057** | 13.372 |
| ADA | 2.360 | 1.373 | 0.515 | 13.128 | 0.021 | 2.116 | 0.059 | 13.464 |
| FOMA | 1.471 | 0.816 | 0.512 | 12.894 | **0.013** | **1.262** | 0.058 | 14.614 |
| **CEMS** | **1.455** | **0.809** | **0.507** | **12.807** | 0.014 | 1.293 | 0.058 | **13.353** |

in Fig. 1, middle left. Similarly, restricting CEMS to a first-order approximation presents a similar behavior (Fig. 1, middle right). Finally, our second-order CEMS method samples the manifold well, even near high curvature areas (Fig. 1, right).

## 5.2 IN-DISTRIBUTION GENERALIZATION

In what follows, we consider the in-distribution benchmark that was introduced in (Yao et al., 2022). This benchmark evaluates the performance of various data augmentation techniques in the setting of training on a train set and its augmentations, while testing on a test set that was sampled from the same distribution as the train set. Thus, a strong performance in this benchmark implies that the underlying DA method mimics the train distribution well. Below, we compare CEMS to other recent state-of-the-art (SOTA) approaches, while using the same datasets that were studied in (Yao et al., 2022) and closely replicating their experimental setup.

**Datasets.** We evaluate in-distribution generalization using four datasets. Two of these are tabular datasets: Airfoil Self-Noise (Airfoil) (Brooks et al., 2014), containing aerodynamic and acoustic measurements of airfoil blade sections, and NO2 (Aldrin, 2004), which predicts air pollution levels at specific locations. We also use two time series datasets: Exchange-Rate and Electricity (Lai et al., 2018), where Exchange-Rate includes daily exchange rates of several currencies and Electricity contains measurements of electric power consumption in private households. For a detailed description of these datasets, see App. G.

**Experimental Settings.** We perform a comparative analysis between our method, CEMS, and several established baseline approaches, including the standard empirical risk minimization (ERM) training, Mixup (Zhang et al., 2018), Manifold-Mixup (Verma et al., 2019), C-Mixup (Yao et al., 2022), Anchor Data Augmentation (ADA) (Schneider et al., 2023), and FOMA (Kaufman & Azencot, 2024). The neural networks we trained are the same as considered in (Yao et al., 2022), where a fully connected three layer model was used for tabular datasets, and an LST-Attn (Lai et al., 2018) is utilized for time series data. The evaluation metrics include the root mean square error (RMSE) and mean absolute percentage error (MAPE). Additional details on experimental settings and hyperparameters are available in App. F.

**Results.** We present the in-distribution generalization benchmark results in Tab. 1. The results of all previous methods are reported as they appear in the corresponding original papers. Lower values are preferred either in RMSE or in MAPE. Boldface and underline denote the best and second best approaches, respectively. Remarkably, across all datasets and metrics, CEMS attains the best or second best error measures. In particular, CEMS outperforms other data augmentation strategies on Airfoil and NO2 while being comparable with FOMA on Electricity and Exchange-Rate. We also note that when CEMS is second best, its result is relatively close to the best result. We present the full results including standard deviation measures in App. H.

## 5.3 OUT-OF-DISTRIBUTION

To extend our in-distribution evaluation, we also consider an out-of-distribution benchmark, as was proposed in (Yao et al., 2022). Unlike the in-distribution case, here the test set is sampled from a distribution different from that of the train set. Therefore, excelling in this scenario provides valuable information regarding the generalization capabilities of data augmentation tools. In what follows, we perform a comparison between CEMS and several SOTA methods, while using the same datasets that were studied in (Yao et al., 2022) and closely replicating their experimental setup.

**Datasets.** We leverage five datasets to evaluate the performance of out-of-distribution robustness. 1) RCFashionMNIST (RCF) (Yao et al., 2022) is a synthetic variation of Fashion-MNIST, designed to model sub-population distribution shifts, with the aim of predicting the rotation angle for each object. 2) Communities and Crime (Crime) (Redmond, 2009) is a tabular dataset focused on predicting the total number of violent crimes per 100,000 population, aiming to create a model that generalizes to states not included in the training data. 3) SkillCraft1 Master Table (SkillCraft) (Blair et al., 2013) is a tabular dataset designed to predict the average latency in milliseconds from the onset of perception-action cycles to the first action where "LeagueIndex" is considered as domain information. 4) Drug-Target Interactions (DTI) (Huang et al., 2021) seeks to predict drug-target interactions that are out-of-distribution, using the year as domain data. 5) PovertyMap (Poverty) (Koh et al., 2021) is a satellite image regression dataset created to estimate asset wealth in countries that were not part of the training set. For more details about the datasets, please refer to App. G.

**Experimental Settings.** Similar to Sec. 5.2, we consider the same baseline DA approaches. For metrics, we report the RMSE (where lower values are preferable) for RCF, Crimes, and SkillCraft. In addition, we use $R$ (where higher values are preferable) as the evaluation metric for Poverty and DTI, as was originally proposed in their corresponding papers Koh et al. (2021); Huang et al. (2021). For a fair comparison, we follow the methodology in (Yao et al., 2022), and we train a ResNet-18 on the RCF and Poverty datasets, three-layer fully connected networks on Crimes and SkillCraft, and DeepDTA Öztürk et al. (2018) on DTI. We provide further details on hyperparameters and experiments in App. F.

**Results.** We detail our out-of-distribution benchmark results in Tab. 2. Similarly to the in-distribution setting, the error measures of previous SOTA approaches were taken from the related original papers. We include both the average (Avg.) and worst domain performance metrics. Lower values are preferred in RMSE, and higher values are opted for $R$. We denote in bold and underline the best and second best results, respectively. Our results indicate that CEMS attains strong performance measures, achieving the best results in $6/9$ tests. Further, the rest of the error measures of CEMS are either second best or very close to the second best. We particularly note the SkillCraft test where CEMS improves the second best results by a relative $1\%$ and $8\%$ for the average and worst metrics. The relative improvement is computed via $e_{rel} \cdot 100$, where $e_{rel} = (e - e_{CEMS})/e$, with $e_{CEMS}$ and $e$ denoting the errors of CEMS and the second best approach, respectively. We present the full results including standard deviation measures in App. H.

## 5.4 ABLATION: BASIS COMPUTATION PER POINT VS. PER NEIGHBORHOOD

As mentioned in Sec. 4, given a data point $z$, we construct a neighborhood $N_z$ which can be used to 1) sample a point $\tilde{z}$ near $z$ 2) sample points $\tilde{N}_z$ near $N_z$. The first option requires estimating a basis for the tangent space for each point in the dataset, whereas the second option estimates a single basis for the the entire batch of points $N_z$. In practice, the first option requires significantly more SVD calculations, determined by the batch size $b$. For large datasets, using the first option becomes very time consuming. In Tab. 3, we compare between Option 1 ($CEMS_p$), where $p$ stands for point and Option 2 (CEMS), considering specifically the smaller datasets. Based on these results, we find that $CEMS_p$ achieves error measures similar to CEMS. However, the computational complexity of $CEMS_p$ is much higher, and thus we advocate the batch-wise computation as suggested in CEMS.

Table 2: Comparison of out-of-distribution robustness. Bold values indicate the best results, while underlined values represent the second best. We present the average RMSE across domains as well as the "worst within-domain" RMSE from three different seeds. For the DTI and Poverty datasets, we provide the average $R$ and the "worst within-domain" $R$. Complete results, including standard deviation, can be found in App H.

| | RCF (RMSE) | Crimes (RMSE) | | SkillCraft (RMSE) | | DTI ($R$) | | Poverty ($R$) | |
|---|---|---|---|---|---|---|---|---|---|
| | Avg. ↓ | Avg. ↓ | Worst ↓ | Avg. ↓ | Worst ↓ | Avg. ↑ | Worst ↑ | Avg. ↑ | Worst ↑ |
| ERM | 0.164 | 0.136 | 0.170 | 6.147 | 7.906 | 0.483 | 0.439 | 0.80 | 0.50 |
| Mixup | 0.159 | 0.134 | 0.168 | 6.460 | 9.834 | 0.459 | 0.424 | **0.81** | 0.46 |
| ManiMixup | 0.157 | 0.128 | 0.155 | 5.908 | 9.264 | 0.474 | 0.431 | - | - |
| C-Mixup | **0.146** | **0.123** | **0.146** | 5.201 | 7.362 | 0.498 | 0.458 | **0.81** | **0.53** |
| ADA | 0.175 | 0.130 | 0.156 | 5.301 | 6.877 | 0.493 | 0.448 | 0.79 | 0.52 |
| FOMA | 0.159 | 0.128 | 0.158 | - | - | 0.503 | 0.459 | 0.78 | 0.49 |
| **CEMS** | **0.146** | 0.128 | 0.159 | **5.142** | **6.322** | 5.11 | **0.465** | **0.81** | 0.50 |

Table 3: Ablation results for estimating the basis per point in the neighborhood ($\text{CEMS}_p$) vs. estimating it once and re-using the basis for every point $N_z$ (CEMS).

| | Airfoil | | NO2 | | Crimes (RMSE) | | SkillCraft (RMSE) | |
|---|---|---|---|---|---|---|---|---|
| | RMSE | MAPE | RMSE | MAPE | Avg. ↓ | Worst ↓ | Avg. ↓ | Worst ↓ |
| $\text{CEMS}_p$ | 1.462 | **0.783** | **0.503** | **12.759** | 0.130 | **0.157** | **5.026** | 8.063 |
| CEMS | **1.455** | 0.809 | 0.507 | 12.807 | **0.128** | 0.159 | 5.142 | **6.322** |

# 6 CONCLUSIONS

This work introduces CEMS, a novel data augmentation method tailored specifically for regression problems, framed within the context of manifold learning. By leveraging second-order manifold approximations, CEMS captures the underlying curvature and structure of the data more accurately than previous first-order methods. Our extensive evaluation across nine diverse benchmark datasets, spanning both in-distribution and out-of-distribution tasks, shows that CEMS achieves competitive performance compared to SOTA techniques, often surpassing them in challenging settings. The main contributions of this work are threefold: (1) extend the view of DA for regression as a manifold learning problem, thereby providing a principled foundation and practical tools; (2) proposing CEMS, a fully differentiable, data-driven, and domain-independent second-order augmentation method; and (3) empirically validating CEMS across a variety of regression scenarios, showing its potential to serve as a robust and effective regularization technique for models predicting continuous values. Our results suggest that higher-order manifold sampling approaches hold promise for improving the generalization of regression models, especially in scenarios with limited data.

One limitation of CEMS is that the linear system in Eq. 9 is might be underdetermined for data sets with a large intrinsic dimension $d$. In practice, the number of neighbors has to be $\mathcal{O}(d^2)$, for an overdetermined system. Our implementation sets the number of neighbors to be a constant size and thus it is independent of $d$. While this requirement is reasonable for low $d$ values, it can become expensive for large $d$. This can be resolved by regularizing the linear system via, e.g., ridge regression. Another limitation is related to the SVD computation, where CEMS needs at least $d$ columns. This may require a full SVD calculation, demanding $\mathcal{O}(bD^2)$ memory, where $b$ is the batch size and $D$ is the extrinsic dimension, which may be impractical for datasets with many features. A potential solution is to consider a different intrinsic dimension $\tilde{d}$, such that $\tilde{d} < d$. In future work, we plan to investigate these ideas, and, in addition, we plan to explore extensions of CEMS to include adaptive strategies for dynamically selecting the appropriate order of approximation based on local data properties. By pushing the boundaries of data augmentation for regression, we hope to pave the way for more robust and versatile learning systems capable of tackling complex, real-world prediction tasks.

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

## A  CURVATURE-AWARE MANIFOLD LEARNING

Given a train set $Z = \{z^1, \cdots, z^N\}$, we parameterize the data manifold around a point $z \in Z$ that we map to $u = 0$, resulting in the following truncated Maclaurin series for a nearby point $u_j$:

$$g^\alpha(u_j) = u_j^T \nabla g^\alpha + \frac{1}{2} u_j^T H^\alpha u_j + \mathcal{O}(|u_j|_2^3), \quad \alpha = 1, \ldots, D - d. \tag{8}$$

In order to estimate the gradient and Hessian of the embedding mapping $g^\alpha$, we build a set of linear equations that solves Eq. 8. Particularly, we approximate $g^\alpha$ by solving the system $G = \Psi X$, where $X$ holds the unknown elements of the gradients $\nabla g^\alpha$ and the Hessians $H^\alpha$, for every $\alpha$. We define $g^\alpha = [g^\alpha(u_1), \cdots, g^\alpha(u_k)]^T \in \mathbb{R}^k$, where $u_j$ are points in the neighborhood of $z := f(u)$, and $G = [g^1, \cdots, g^{D-d}]$. The point $u$ and points $\{u_j\}$ are associated with the train set $Z$ under a natural orthogonal transformation. The local natural orthogonal coordinates are a set of coordinates that are defined at a specific point $u$ of the manifold. They are constructed by finding a basis for the tangent space and normal space at a point $u$ by applying principal component analysis on the neighborhood $N_z = \{z_j\}_{j=1}^k$. Namely, the first $d$ coordinates (associated with the most significant modes, i.e., largest singular values) represent the tangent space, and the rest represent the normal space. Then, we define $\Psi = [\Psi_1, \cdots, \Psi_k]$, stacking $\Psi_j$ in rows, where $\Psi_j$ is given via

$$\Psi_j = \left[ u_j^1, \cdots, u_j^d, \left(u_j^1\right)^2, \cdots, \left(u_j^d\right)^2, \left(u_j^1 \times u_j^2\right), \cdots, \left(u_j^{d-1} \times u_j^d\right) \right],$$

and

$$X^\alpha = \left[ \nabla g^{\alpha 1}, \cdots, \nabla g^{\alpha d}, H^{1,1}, \cdots, H^{d,d}, H^{1,2}, \cdots, H^{\alpha d-1, d} \right]^T,$$

with $X = [X^1, \cdots, X^{D-d}]$. The set of linear equations

$$G = \Psi X, \tag{9}$$

is solved by using the least square estimation given $X = \Psi^\dagger G$. In practice, we estimate only the upper triangular part of $H^\alpha$ since it is a symmetric matrix. We refer the reader for a more comprehensive and detailed treatment in (Li, 2018).

## B  COMPUTATIONAL COST

**General analysis of $n$-th order embedding maps:**   To estimate the $n$-th order Taylor approximation of the embedding $g$, we need to find all the partial derivatives up to and including order $n$. The gradient $\nabla$ vector is composed of $d$ first-order partial derivatives, the Hessian matrix is composed of $d^2$ second-order partial derivatives. Third-order and higher partial derivatives are represented using mathematical objects called tensors. In general, the number of partial derivatives of a multi-input function grows exponentially with the order.

The order $n$ of the partial derivatives determines the number of unknowns $X$ and the size of the matrix $\Psi$ which used to solve Eq. 9 via least squares. The number of columns in the matrix $\Psi$ is $\sum_{i=1}^n d^i = \mathcal{O}(d^n)$, and thus, the dimensions of the matrix $\Psi$ are $k \times d^n$. It is preferred to solve an over-determined set of equations such that the number of neighbors $k > d^n$, which is memory and computationally expensive. For small values of $d$ and $n$ it is feasible to solve Eq. 9 in a run time complexity of $\mathcal{O}(kd^{2n})$. For larger values of $n$, it becomes extremely computationally and memory expensive to achieve $k \geq d^n$, therefore, we can assume that $k < d^n$.

There are two computationally expensive operations used to estimate the $n$-th order approximation of the manifold, SVD and least squares. The matrix on which we perform SVD is of shape $N_z \in \mathbb{R}^{k \times D}$ where $k$ represents the number of neighbors and $D$ is the extrinsic dimension. Thus, the run time complexity of the SVD operation per batch is $\mathcal{O}(\min(k^2 D, kD^2))$. The complexity of least squares is determined by the dimensions of the matrix $\Psi \in \mathbb{R}^{k \times \mathcal{O}(d^n)}$ resulting in $\mathcal{O}(kd^{2n})$. If we wish to solve an over-determined system of equations, we need to set $k > d^n$ e.g., $k = 2d^n$ resulting in a run time of $\mathcal{O}(\min(d^{2n}D, d^n D^2))$ for SVD and $\mathcal{O}(d^n d^{2n}) = \mathcal{O}(d^{3n})$ for least squares.

**Computational analysis of CEMS.**   Given a batch of data $A \in \mathbb{R}^{b \times D}$ where $b$ is the batch size and $D$ is the ambient dimension, our analysis considers the point-wise and batch-wise settings.

In the point-wise case, we construct a matrix $N_a \in \mathbb{R}^{b \times D}$ for every sample $a \in \mathbb{R}^D$ in the batch, containing its $b$ closest neighbors, where the number of neighbors is fixed as the batch size. This practical choice decouples the computational complexity from the intrinsic dimension $d$. On each matrix $N_a$, we perform SVD at the complexity of $\mathcal{O}(\min(bD^2, Db^2))$. We then solve the set of $b$ equations with $l = d \times (d+1)/2$ variables (representing the unknowns in the gradient $\nabla$ and Hessian $H$) at a complexity of $\mathcal{O}(b \times l^2) = \mathcal{O}(b \times d^4)$. In practice, the batch size is small and constant which leads to an underdetermined system that can be solved used Ridgr-Regreession at a complexity of $\mathcal{O}(b^2 \times l) = \mathcal{O}(b^2 \times d^2)$. The total complexity per point is therefore $\mathcal{O}(\min(bD^2, Db^2)) + \mathcal{O}(b^2 \times d^2)$. Since the batch size $b$ is usually smaller than the ambient dimension $D$, the total complexity can be revised as $\mathcal{O}(b^2(D + d^2))$. Under the manifold hypothesis, we assume that $d << D$ and thus $d \in \mathcal{O}(D^2)$, resulting in the following complexity $\mathcal{O}(b^2 D)$ for a single point and $\mathcal{O}(b^3 D)$ for the entire batch. Our analysis reveals that under our assumptions, the complexity is proportional to the ambient dimension $D$.

In the batch-wise setting, the entire batch $A \in \mathbb{R}^{b \times D}$ is processed collectively. We compute the SVD of $A$ at a complexity of $\mathcal{O}(\min(bD^2, Db^2))$. The subsequent step involves solving $b$ equations with $l = d \times (d+1)/2$ variables at a complexity of $\mathcal{O}(b \times l^2)$. As in the point-wise case The total computational complexity of CEMS for the batch-wise setting for a single batch is $\mathcal{O}(b^2 D)$

**Run time comparison** In Table 4, we compare the total run time of the training process in seconds to provide an estimate for the empirical computational cost of CEMS and competing methods. The results are obtained with a single `RTX3090` GPU. For each data set, all the methods were estimated using the same parameters (e.g., batch size, number of epochs) for a fair comparison. It is evident from the results that the empirical run time of CEMS is o par with competing methods and does not require a large overhead.

Table 4: Training times comparison (in seconds).

|          | AIRFOIL | NO2  | RCF  | DTI  |
|----------|---------|------|------|------|
| ERM      | 3.84    | 1.01 | 172  | 653  |
| C-MIXUP  | 11.64   | 2.04 | 1700 | 1064 |
| ADA      | 8.72    | 3.22 | 465  | 3519 |
| FOMA     | 7.11    | 1.85 | 364  | 1095 |
| CEMS     | 12.2    | 3.04 | 445  | 1317 |

## C  BATCH SELECTION

In our framework, for any data point $(x, y) \in \mathcal{X} \times \mathcal{Y}$, we seek to generate training samples that lie near the manifold $\mathcal{M}$, which represents the true data distribution $\mathcal{P}$. Since directly sampling from $\mathcal{M}$ is impossible without knowing its structure, we instead approximate it locally using the tangent plane $\mathcal{T}(x, y)$ at point $(x, y)$.

To construct this tangent plane approximation, we require a neighborhood set $N_z$ around $z$. We compute the tangent plane using Singular Value Decomposition (`SVD`) on these neighboring points. The accuracy of this approximation heavily depends on how close the points in $N_z$ are to $z$.

To implement this approach, we explored three distinct batch construction methods:

1. Random batch selection (random)

Table 5: Results for different batch selection methods `CEMS`.

| Dataset        | Airfoil↓ | NO2↓   | SkillCraft↓ | RCF↓   | DTI↑   |
|----------------|----------|--------|-------------|--------|--------|
| CEMS - knn     | 1.455    | 0.507  | 5.142       | **0.146** | **0.511** |
| CEMS - knnp    | 1.441    | **0.506** | **4.941**   | 0.173  | 0.509  |
| CEMS - random  | **1.435** | 0.506  | 5.155       | 0.162  | 0.491  |

2. k-Nearest Neighbors batch selection (knn), where points are grouped based on their Euclidean distances in $Z$

3. k-Nearest Neighbors Probability batch selection (knnp), where points are sampled with probabilities inversely proportional to their distance from the original point, ensuring closer points have higher sampling probabilities

Our hypothesis was that both proximity-based and probability-based batch selection methods would generate samples closer to the data manifold, thereby improving model performance. We first trained models using the proximity-based batch selection method and selected the best-performing model based on the validation set. Using the optimal parameters found from this model, we then trained two additional models using random batch selection and knnp methods for fair comparison. The experimental results, presented in Table 5, demonstrate the relative performance of these three approaches under identical parameter settings. The results reveal interesting patterns across different batch selection methods. While no single method dominates across all datasets, each approach shows strengths in specific scenarios. The knnp method demonstrates superior performance on the SkillCraft dataset and matches the best performance on NO2. The knn approach excels in both RCF and DTI datasets, showing particular strength in structured data scenarios. Interestingly, random batch selection remains competitive, achieving the best performance on Airfoil and matching the best result on NO2. These results suggest that the effectiveness of batch selection methods may be dataset-dependent, highlighting the importance of considering data characteristics when choosing a batch selection strategy.

## C.1 NEIGHBORHOOD CONSTRUCTION

The validity of CEMS relies on three key theoretical foundations. First, our neighborhood construction approach balances computational efficiency with geometric fidelity by sharing neighborhoods across points in close proximity. While this might appear to reduce sampling diversity, it actually preserves manifold structure because points that are close in the normalized input-output space typically share similar geometric properties. The shared neighborhood assumption is particularly valid because we normalize both input $X$ and output $Y$ features to the same range, ensuring that proximity in the combined space meaningfully reflects similarity in both domains.

The reliability of our construction is maintained even in regions of high output diversity through our careful treatment of the input-output space. For a point $z_i = [x_i, y_i]$, its neighborhood $\mathcal{N}_z$ is constructed considering distances in both input and output spaces simultaneously, naturally limiting the diversity of outputs within each neighborhood. This approach ensures that points sharing a neighborhood basis have similar geometric properties, maintaining the validity of our second-order approximation.

The stochastic nature of mini-batch training provides an additional beneficial property for CEMS. As different batches are sampled each epoch, the method naturally explores varying neighborhoods and their associated tangent spaces. This dynamic sampling process enables CEMS to build a comprehensive representation of the manifold's local geometry, adapting to variations in data density across different regions. The continuous exploration of diverse local structures throughout training enhances the method's ability to capture the full geometric complexity of the underlying manifold.

The local-Euclidean structure of CEMS is supported by two complementary mechanisms. First, the second-order approximation naturally captures local curvature through the Hessian term, enabling accurate representation of nonlinear geometries. Second, our stochastic batch sampling strategy ensures exposure to different neighborhoods and tangent spaces throughout training. The projection of points onto the tangent space at $\mu$ (the neighborhood mean) maintains validity through the second-order terms in our Taylor expansion, which account for the primary nonlinearities within each neighborhood. This approach is particularly robust because the neighborhood size adapts with the batch size, preserving accurate local approximations even in regions of high curvature.

The empirical success of this construction is demonstrated in our ablation studies (Table 3), where we compare point-wise basis computation ($CEMS_p$) with our more efficient shared neighborhood approach (CEMS). The comparable performance across multiple datasets validates our theoretical assumptions about the effectiveness of shared geometric information within local neighborhoods.

## C.2 Local vs. Global Sampling

It is important to note that CEMS focuses on local sampling within neighborhoods where the second-order approximation is valid. While alternative approaches based on geodesics can enable global sampling along the manifold, they typically incur significantly higher computational costs. Our local approach strikes a balance between sampling accuracy and computational efficiency, making it particularly well-suited for data augmentation during training.

The theoretical guarantees provided by Theorem 4.1 hold within the neighborhood where our Taylor approximations are valid. This aligns with the manifold hypothesis, which posits that real data typically lies on or near a lower-dimensional manifold with locally Euclidean structure (Goodfellow, 2016; Belkin & Niyogi, 2003). By focusing on accurate local sampling, CEMS can effectively augment the training data while maintaining the essential geometric structure of the underlying manifold.

## D  Geometric Properties Effect on CEMS

To evaluate the impact of curvature on CEMS for regression tasks, we generated a synthetic dataset where features lie on a manifold of constant scalar curvature. The manifold is defined as a hypersphere with a constant scalar curvature. Data points are sampled uniformly on the hypersphere using normalized random directions and embedded into a higher-dimensional ambient space through a deterministic projection matrix to preserve the manifold structure. The regression target $Y$ is computed as a non-linear function of the intrinsic coordinates, $Y = \sin\left(\sum X_{\text{intrinsic}}\right)$, introducing a smooth dependency on the features. The features and targets are normalized to lie within the range $[0, 1]$ using min-max scaling. This setup enables the systematic study of the effects of curvature of CEMS on regression model performance. In Fig. 3 the graph illustrate the relative improvement in RMSE between the CEMS and baseline ERM for regression tasks across varying scalar curvatures of the data manifold. The graph plots relative improvement against scalar curvature, highlighting that CEMS provides minimal advantage for nearly flat manifolds with low curvature but exhibits increasing improvement as the curvature grows. At higher curvatures (e.g., 16–64), CEMS demonstrates substantial gains, reflecting its ability to exploit geometric information in highly curved spaces. The hyperparameters of CEMS were not fine-tuned for this experiment and remained consistent across all intrinsic dimensions and curvature values. This likely explains why, in some cases, CEMS does not achieve better performance than ERM.

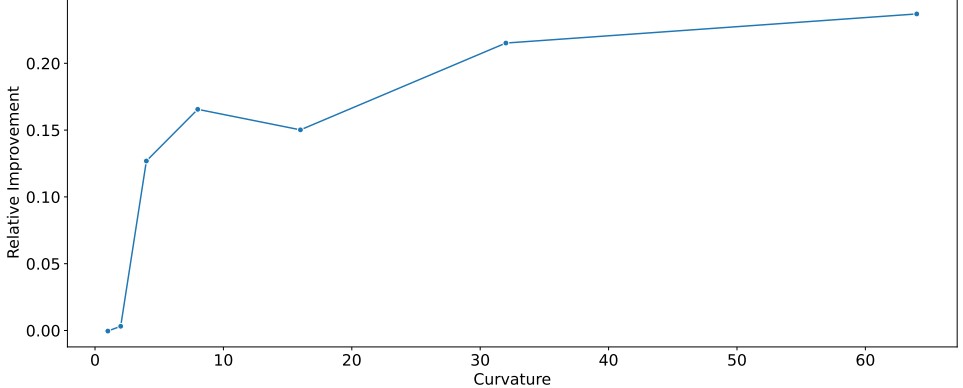

Figure 3: Illustration of the relative improvement in RMSE of CEMS over ERM. The graph demonstrates the effect of curvature, indicating minimal gains for nearly flat manifolds but substantial improvements for highly curved manifolds. These results emphasize the influence of data geometry on the performance of CEMS relative to ERM.

## E  Adaptation to batches

Below we provide the algorithm for the batched version:

---

**Algorithm 2** Curvature Enhanced Manifold Sampling (CEMS-batch)

---

**Require:** Training data $Z = \{z^i = [x^i, y^i]\}_{i=1}^N$. A sample $z \in Z$

1:   Find K-nearest neighbors $N_z = \{z_j\}_{j=1}^k \cup \{z = z_0\}$ of $z$
2:   Find an orthonormal basis $B_u$ that spans $N_z - \mu_{N_z}$
3:   Project every $z_j - \mu_{N_z}$ to the local orthonormal coordinates:
4:       $u_j = B_{\mathcal{T}_u}^T \cdot (z_j - \mu_{N_z}), g_j = B_{\mathrm{N}_u}^T \cdot (z_j - \mu_{N_z})$
5:   For each $l = 1, \ldots, k$ construct:
6:       $U_z^l = \{u_j - u_l\}_{j \neq l}^k$ and $G_z^l = \{g_j - g_l\}_{j \neq l}^k$
7:       Construct $G$ and $\Psi$ as in Eq. 9
8:       Solve $\Psi A = G$
9:       Extract $\nabla g(z)$ and $H(z)$ from $A$
10:      Sample a point $\eta$ near $u_l$: $\eta \sim \mathcal{N}(u_l, \sigma I_d)$
11:      Calculate $g(\eta_l) = g(u_l) + (\eta_l - u_l)^T \nabla g + \frac{1}{2}(\eta_l - u_l)^T H (\eta_l - u_l)$
12:      Un-project $\eta_l$ back to the original coordinates:
13:          $z_{\eta_l} := f(\eta_l) = B_u \cdot [\eta_l, g(\eta_l)] + \mu_{N_z}$
14: **return** $z_\eta = \{z_{\eta_l}\}_{l=1}^k$

---

## F   Hyperparameters

We present the hyperparameters for each dataset in Table 6. In our main results, we apply our method to the input space or the latent space, and we report the configuration with the best performance. All hyperparameters were selected through cross-validation and evaluated on the validation set. Some hyperparameters, such as architecture and optimizer, are not included in the tables since they remained unchanged and were used as specified in previous works Yao et al. (2022); Schneider et al. (2023).

Table 6: Hyperparameter choices for the experiments using CEMS.

| Dataset | Airfoil | NO2 | Exchange-Rate | Electricity | RCF | Crimes | SkillCraft | PovertyMap | DTI |
|---|---|---|---|---|---|---|---|---|---|
| Learning rate | $1e^{-3}$ | $1e^{-3}$ | $1e^{-3}$ | $5e^{-4}$ | $1e^{-4}$ | $1e^{-3}$ | $1e^{-3}$ | $5e^{-3}$ | $1e^{-4}$ |
| Batch size | 16 | 32 | 32 | 32 | 64 | 16 | 16 | 32 | 32 |
| Input/Manifold | manifold | input | input | manifold | manifold | manifold | input | input | input |
| Epochs | 700 | 100 | 200 | 100 | 50 | 100 | 100 | 50 | 60 |
| $\sigma$ | $1e^{-4}$ | 0.2 | 0.1 | $1e^{-3}$ | 0.01 | 0.3 | 0.2 | 0.1 | $1e^{-3}$ |

## G   Dataset Description

This section provides detailed descriptions of the datasets used in our experiments.

**Airfoil Self-Noise (Brooks et al., 2014).**   This dataset contains aerodynamic and acoustic test data for various NACA 0012 airfoils, recorded at different wind tunnel speeds and angles of attack. Each data point includes five features: frequency, angle of attack, chord length, free-stream velocity, and suction side displacement thickness, with the label representing the scaled sound pressure level. Input features are normalized using min-max normalization. The dataset is divided into 1003 training examples, 300 validation examples, and 200 test examples as noted in Hwang & Whang (2021).

**NO2 (Aldrin, 2004).**   The NO2 dataset examines the relationship between air pollution near a road and traffic volume along with meteorological variables. Each input consists of seven features: the logarithm of the number of cars per hour, temperature at 2 meters above ground, wind speed, temperature difference between 25 and 2 meters above ground, wind direction, hour of the day, and the day number since October 1, 2001. The response variable is the hourly logarithm of NO2 concentration measured in Oslo from October 2001 to August 2003. The dataset is split into 200 training examples, 200 validation examples, and 100 test examples as in Hwang & Whang (2021).

**Exchange-Rate (Lai et al., 2018).**   This time series dataset includes daily exchange rates for eight countries (Australia, Britain, Canada, Switzerland, China, Japan, New Zealand, and Singapore) from

1990 to 2016, totaling 7,588 observations with daily frequency. A sliding window size of 168 days is applied, resulting in an input dimension of $168 \times 8$ and a label dimension of $1 \times 8$ data points. The dataset is partitioned into training (60%), validation (20%), and test (20%) sets in chronological order as described in Lai et al. (2018).

**Electricity (Lai et al., 2018).**   This dataset contains hourly electricity consumption data from 321 clients, recorded every 15 minutes from 2012 to 2014, totaling 26,304 observations. A sliding window size of 168 is used, resulting in an input dimension of $168 \times 321$ and a label dimension of $1 \times 321$. The dataset is divided into training, validation, and test sets following a methodology similar to that used for the Exchange-Rate dataset.

**RCF (Yao et al., 2022).**   The RCF-MNIST (Rotated-Colored-Fashion) dataset features images with specific color and rotation attributes. Images are colored using RGB vectors based on the rotation angle $g \in [0, 1]$. In the training set, 80% of images are colored with $[g, 0, 1 - g]$, and 20% with $[1 - g, 0, g]$, creating a spurious correlation between color and label.

**PovertyMap (Koh et al., 2021).**   Part of the WILDS benchmark (Koh et al., 2021), this dataset consists of satellite images from 23 African countries used to predict village-level asset wealth. Each input is a $224 \times 22$ multispectral LandSat image with 8 channels, and the label is the real-valued asset wealth index. The dataset is divided into 5 cross-validation folds with disjoint countries to facilitate the out-of-distribution setting, following the methodology in Koh et al. (2021).

**Crime (Redmond, 2009).**   The Communities And Crimes dataset merges socio-economic data from the 1990 US Census, law enforcement data from the 1990 US LEMAS survey, and crime data from the 1995 FBI UCR. It includes 122 attributes related to crime, such as median family income and percentage of officers in drug units. The target is the per capita violent crime rate. Numeric features are normalized to a range of 0.00 to 1.00, and missing values are imputed. The dataset is divided into training (1,390), validation (231), and test (373) sets, with 31, 6, and 9 disjoint domains, respectively.

**SkillCraft (Blair et al., 2013).**   The SkillCraft dataset from UCI consists of video game telemetry data from real-time strategy (RTS) games, focusing on player expertise development. Each input includes 17 player-related parameters, such as cognition-action-cycle variables and hotkey usage, while the label is the action latency. Missing data are filled by mean padding. The dataset is divided into training (1,878), validation (806), and test (711) sets with 4, 1, and 3 disjoint domains, respectively.

**DTI (Huang et al., 2021).**   The Drug-Target Interactions dataset aims to predict the binding activity score between small molecules and target proteins. Input features include one-hot vectors for drugs and target proteins, and the output is the binding activity score. Training and validation data are from 2013 to 2018, while the test data spans 2019 to 2020. The "Year" attribute serves as domain information.

## H    RESULTS WITH STANDARD DEVIATION

In Table 7 we report the full results of in-distribution generalization and in Table 8 we report the full results of out-of-distribution robustness.

Table 7: Full results for in-distribution generalization. Standard deviations are calculated over 3 seeds.

|  | Airfoil | | NO2 | |
|---|---|---|---|---|
|  | RMSE | MAPE | RMSE | MAPE |
| ERM | $2.901 \pm 0.067$ | $1.753 \pm 0.078$ | $0.537 \pm 0.005$ | $13.615 \pm 0.165$ |
| Mixup | $3.730 \pm 0.190$ | $2.327 \pm 0.159$ | $0.528 \pm 0.005$ | $13.534 \pm 0.125$ |
| Mani Mixup | $3.063 \pm 0.113$ | $1.842 \pm 0.114$ | $0.522 \pm 0.008$ | $13.357 \pm 0.214$ |
| C-Mixup | $2.717 \pm 0.067$ | $1.610 \pm 0.085$ | $\underline{0.509 \pm 0.006}$ | $12.998 \pm 0.271$ |
| ADA | $2.360 \pm 0.133$ | $1.373 \pm 0.056$ | $0.515 \pm 0.007$ | $13.128 \pm 0.147$ |
| FOMA | $\underline{1.471 \pm 0.047}$ | $\underline{0.816 \pm 0.008}$ | $0.512 \pm 0.008$ | $\underline{12.894 \pm 0.217}$ |
| **CEMS** | $\mathbf{1.455 \pm 0.119}$ | $\mathbf{0.809 \pm 0.050}$ | $\mathbf{0.507 \pm 0.003}$ | $\mathbf{12.807 \pm 0.044}$ |

|  | Exchange-Rate | | Electricity | |
|---|---|---|---|---|
|  | RMSE | MAPE | RMSE | MAPE |
| ERM | $0.023 \pm 0.003$ | $2.423 \pm 0.365$ | $\underline{0.058 \pm 0.001}$ | $13.861 \pm 0.152$ |
| Mixup | $0.023 \pm 0.002$ | $2.441 \pm 0.286$ | $\underline{0.058 \pm 0.000}$ | $14.306 \pm 0.048$ |
| Mani Mixup | $0.024 \pm 0.004$ | $2.475 \pm 0.346$ | $\underline{0.058 \pm 0.000}$ | $14.556 \pm 0.057$ |
| C-Mixup | $0.020 \pm 0.001$ | $2.041 \pm 0.134$ | $\mathbf{0.057 \pm 0.001}$ | $13.372 \pm 0.106$ |
| ADA | $0.021 \pm 0.006$ | $2.116 \pm 0.689$ | $0.059 \pm 0.001$ | $13.464 \pm 0.296$ |
| FOMA | $\mathbf{0.013 \pm 0.000}$ | $\mathbf{1.262 \pm 0.037}$ | $\underline{0.058 \pm 0.000}$ | $14.653 \pm 0.166$ |
| **CEMS** | $\underline{0.014 \pm 0.001}$ | $\underline{1.269 \pm 0.062}$ | $\underline{0.058 \pm 0.000}$ | $\mathbf{13.353 \pm 0.217}$ |

Table 8: Full results for out-of-distribution robustness. Standard deviations are derived from a 5-fold data split in PovertyMap and or calculated over 3 seeds for other datasets.

|  | RCF (RMSE) | Crimes (RMSE) | | SkillCraft (RMSE) | |
|---|---|---|---|---|---|
|  | Avg. ↓ | Avg. ↓ | Worst ↓ | Avg. ↓ | Worst ↓ |
| ERM | $0.164 \pm 0.007$ | $0.136 \pm 0.006$ | $0.170 \pm 0.007$ | $6.147 \pm 0.407$ | $7.906 \pm 0.322$ |
| Mixup | $0.159 \pm 0.005$ | $0.134 \pm 0.003$ | $0.168 \pm 0.017$ | $6.461 \pm 0.426$ | $9.834 \pm 0.942$ |
| ManiMixup | $0.157 \pm 0.021$ | $0.128 \pm 0.003$ | $0.155 \pm 0.009$ | $5.908 \pm 0.344$ | $9.264 \pm 1.012$ |
| C-Mixup | $\mathbf{0.146 \pm 0.005}$ | $\mathbf{0.123 \pm 0.000}$ | $\mathbf{0.146 \pm 0.002}$ | $\underline{5.201 \pm 0.059}$ | $7.362 \pm 0.244$ |
| ADA | $0.163 \pm 0.014$ | $0.130 \pm 0.003$ | $0.156 \pm 0.007$ | $5.301 \pm 0.182$ | $\underline{6.877 \pm 1.267}$ |
| FOMA | $0.159 \pm 0.010$ | $\underline{0.128 \pm 0.004}$ | $0.158 \pm 0.002$ | - | - |
| **CEMS** | $\mathbf{0.146 \pm 0.002}$ | $\underline{0.128 \pm 0.001}$ | $0.159 \pm 0.004$ | $\mathbf{5.142 \pm 0.143}$ | $\mathbf{6.322 \pm 0.191}$ |

|  | DTI ($R$) | | Poverty ($R$) | |
|---|---|---|---|---|
|  | Avg. ↑ | Worst ↑ | Avg. ↑ | Worst ↑ |
| ERM | $0.483 \pm 0.008$ | $0.439 \pm 0.016$ | $\underline{0.80 \pm 0.04}$ | $0.50 \pm 0.07$ |
| Mixup | $0.459 \pm 0.013$ | $0.424 \pm 0.003$ | $\mathbf{0.81 \pm 0.04}$ | $0.46 \pm 0.03$ |
| ManiMixup | $0.474 \pm 0.004$ | $0.431 \pm 0.009$ | - | - |
| C-Mixup | $0.498 \pm 0.008$ | $0.458 \pm 0.004$ | $\mathbf{0.81 \pm 0.03}$ | $\mathbf{0.53 \pm 0.07}$ |
| ADA | $0.493 \pm 0.010$ | $0.448 \pm 0.009$ | $0.79 \pm 0.03$ | $\underline{0.52 \pm 0.06}$ |
| FOMA | $\underline{0.503 \pm 0.008}$ | $\underline{0.459 \pm 0.010}$ | $0.77 \pm 0.03$ | $0.49 \pm 0.05$ |
| **CEMS** | $\mathbf{5.110 \pm 0.005}$ | $\mathbf{0.465 \pm 0.004}$ | $\mathbf{0.81 \pm 0.05}$ | $0.50 \pm 0.07$ |

