# OpenReview forum: "Curvature Enhanced Manifold Sampling"
_ICLR.cc/2025/Conference — Submitted to ICLR 2025_

### Official Review · Reviewer_ehPU · 2024-10-29

**Soundness:** 2
**Presentation:** 2
**Contribution:** 3
**Rating:** 8
**Confidence:** 3

**Summary:**

This submission focuses on improving the generalization ability of the model with data augmentation (DA). Specifically, the sample generation framework for extending the training set is considered. For existing advances in DA, e.g., linear interpolation in Euclidean space under MixUp framework and manifold sampling under the first-order approximation of data manifold, a common challenge is to ensure the validity of sampling for continuous output spaces and highly nonlinear structure, e.g., regression. This work proposes a noise robustness framework on the manifold, which generates noise-perturbed samples on the tangent space via second-order approximation, and projects them back to the manifold via retraction. A fast computation algorithm is developed to reduce the complexity of the second-order approximation with gradients and Hessians. Compared with SOTA in-distribution/out-of-distribution generalization method, the proposed method achieves superior performance in different benchmarks.

**Strengths:**

+ The motivation is clear, where the main challenge is to characterize a complex manifold structure that lies in the complex output space.

+ The method is technically sound, which employs the second-order approximation formulation with gradients and Hessian to sample new points in the tangent spaces.

+ The superior experiment results over SOTA DA methods on different generalization learning tasks.

**Weaknesses:**

W1. Methodology. The fast computation with the designed neighborhood extraction strategy seems to be sensitive to the data structure where further justifications are needed.

W2. Clarity of Presentation. Though the sampling framework is clearly presented, it seems that the learning process with the generated samples is not clearly stated.

W3. Significance. Though the steps of existing first-order method FOMA is discussed, the difference between FOMA and proposed second-order method CEMS is not formally discussed, i.e., there seems no rigorous results or in-depth analysis on the fundamental advantages of DA with second-order approximation from mathematical aspects.

**Questions:**

Q1. As far as I understand the proposed neighborhood extraction strategy, once neighbor set $N_z$ is computed for $z$, this set will be shared by all points in $N_z$; consequently, the manifold sampling operation, e.g., projection for tangent space and retraction for generation, will be identical for points in $N_z$. 1) As such a partition of $N_z$ seems to boil down to a clustering process, will the quality of generated samples be sensitive to $N_z$? 2) Could such a partition process be replaced by more reliable clustering algorithms?

Q2. Following Q1, if $N_z$ contains the sample with high diversity w.r.t outputs, the generation process seems could be error-prone. For example, considering the classification scenarios with discrete output space, if $N_z$ contains multiple inter-class samples while they still share the same projection basis, it could induce discriminability degradation on patterns.

Q3. As mentioned in the motivation, the primary goal is to ensure the generation quality in a highly nonlinear space. According to the fast computation pipeline in lines 278-286, all points in $N_z$ are projected onto the tangent space in $\mu$, i.e., the mean of $N_z$. If the structure near $N_z$ is highly nonlinear, how to ensure the generation and local structure characterization of points $u_l \in N_z$ are still valid?

Q4. Following Q3, for the generation process in Line 285, it seems that the generation points are distributed around $\mu$ while not $u_l$, i.e., it seems to be equivalent to the random sample $\eta \sim \mathcal{N}(0,\sigma I_d)$ for $N$ times ($N$ is the number of samples) and then substitute them into Eq. (2). Compared with the point-wise computation on each $u_l$, the fast computation saves time while losing the sample information. Moreover, it seems that there would be only a few real samples (i.e., ignoring the noise $\mu$ in the sampling process) generated samples (i.e., the means $\mu$ over different $N_z$). In-depth analysis is highly expected.

Q5. How to justify the advantage of second-order approximation for sampling from a mathematical aspect? Specifically, compared with the first-order approximation, how can the proposed method ensure better quality? Though empirical observation is provided, i.e., Fig. 1, it would be apricated to justify them from fundamental aspects, e.g., the differences between employing Eq. (2) (second-order) and lines 294-304 (first-order).

Q6. The essential of CEMS is ensuring the robustness w.r.t noise on manifold space, while the term 'manifold sampling' in tile could be inappropriate since it seems that CEMS can only guarantee sampling in a local area with the Taylor expansion. Thus, it would be interesting to discuss the CEMS with the geodesic-based sampling methods that can generate unseen samples along the manifold spaces globally.

Q7. Possible typos: DTI metric of CEMS in Tab. 2, where the average value 5.11 is significantly larger than other methods.

---

> ### Author Response · Authors · 2024-11-22
>
> Thank you for your detailed and thoughtful review. We sincerely appreciate your recognition of our work's strengths, including its clear motivation, sound methodology, and superior experimental results. Your insightful comments and questions, particularly on the sensitivity of our neighborhood extraction strategy, the comparison of first- and second-order approximations, and the local versus global sampling scope, provide valuable guidance for improving our presentation and addressing potential limitations. We are grateful for your constructive feedback, which will significantly help us refine our manuscript and enhance its clarity and rigor. Given the opportunity, we will incorporate the following discussions and results into our final revision.
>
> 1. **W1. Methodology. The fast computation with the designed neighborhood extraction strategy seems to be sensitive to the data structure where further justifications are needed.**
>
>    The design of our neighborhood extraction is guided by balancing computational efficiency with the quality of manifold approximation. While this method assumes that the local neighborhood adequately represents the manifold's geometry, we acknowledge that certain data distributions (e.g., those with sparse or irregular sampling) may challenge this assumption. Importantly, we designed a data augmentation approach that makes minimal assumptions about the structure of the data manifold. To achieve this, we rely on general neighborhood approximation techniques, such as k-nearest neighbors, ensuring broad applicability and flexibility.
>    We added an ablation study that explores three distinct batch construction methods: random batch selection, k-Nearest Neighbors (knn) batch selection, and k-Nearest Neighbors Probability (knnp) batch selection. Our hypothesis was that proximity-based and probability-based batch selection methods would generate samples closer to the data manifold, thereby improving model performance. The results suggest that the effectiveness of batch selection methods is indeed dataset-dependent.
>
> 2. **W2. Clarity of Presentation. Though the sampling framework is clearly presented, it seems that the learning process with the generated samples is not clearly stated.**
>
>    To clarify, the generated samples are directly used to train the model, where every sample $z_i$ in the original dataset is used only once during an epoch, and the learning process remains standard. We will revise the manuscript to explicitly outline the training pipeline to detail how the CEMS-generated samples are incorporated during training.
>
> 3. **W3. Significance. Though the steps of existing first-order method FOMA is discussed, the difference between FOMA and proposed second-order method CEMS is not formally discussed, i.e., there seems no rigorous results or in-depth analysis on the fundamental advantages of DA with second-order approximation from mathematical aspects.**
>
>    CEMS leverages second-order sampling from the data manifold, offering a significant improvement over existing methods like FOMA, which rely on first-order approximations. Mathematically, our argument is straightforward: incorporating curvature information, as CEMS does, enables more accurate approximation and sampling of the manifold compared to methods that disregard curvature. This is demonstrated in a toy example in Fig. 1, which highlights the limitations of first-order methods in capturing curved manifolds. Furthermore, our empirical results underscore the critical role of curvature in effective data augmentation for regression. Notably, modern machine learning research often assumes that high-dimensional real-world data resides on curved manifolds [1, 2], further supporting the theoretical foundation and motivation for CEMS. Please also see our response in Q5.

---

> ### Author Response · Authors · 2024-11-22
>
> 4. **Q1. As far as I understand the proposed neighborhood extraction strategy, once neighbor set $N_z$ is computed for $z$, this set will be shared by all points in $N_z$; consequently, the manifold sampling operation, e.g., projection for tangent space and retraction for generation, will be identical for points in $N_z$. 1) As such a partition of $N_z$ seems to boil down to a clustering process, will the quality of generated samples be sensitive to $N_z$? 2) Could such a partition process be replaced by more reliable clustering algorithms?**
>
>    Your understanding is precise. The quality of generated samples is influenced by the neighborhood size density and the manifold's local curvature. Larger neighborhoods may smooth out local variations, while smaller ones risk failing to capture sufficient geometric structure. While clustering methods could theoretically replace neighborhood extraction, they introduce additional complexity and might not align with the manifold hypothesis, which assumes a continuous local structure. Specifically, clustering methods may conflict with the manifold hypothesis because they prioritize partitioning the data into discrete groups, which can disrupt the natural, continuous structure of the manifold. By adhering to local geometric properties, the kNN-based approach in CEMS is better suited to capturing the manifold's inherent continuity and locality, while enabling effective sampling.
>
> 5. **Q2. Following Q1, if $N_z$ contains the sample with high diversity w.r.t outputs, the generation process seems could be error-prone. For example, considering the classification scenarios with discrete output space, if $N_z$ contains multiple inter-class samples while they still share the same projection basis, it could induce discriminability degradation on patterns.**
>
>    We agree that if the shared neighborhood contains samples with high diversity in outputs this could reduce discriminability. The neighborhood of $z_i=[x_i,y_i]$ is constructed from points in close proximity. This construction takes into account the distance between the input $x_i$ and the output $y_i$ ($X$ and $Y$ are normalized to have the same feature range) such that the diversity of the outputs $y_i$ and $y_j$ of $z_i=[x_i,y_i]$ and $z_j=[x_j,y_j]$ (in the same neighborhood), is less pronounced.
>
> 6. **Q3. As mentioned in the motivation, the primary goal is to ensure the generation quality in a highly nonlinear space. According to the fast computation pipeline in lines 278-286, all points in $N_z$ are projected onto the tangent space in $\mu$, i.e., the mean of $N_z$. If the structure near $N_z$ is highly nonlinear, how to ensure the generation and local structure characterization of points $u_l \in N_z$ are still valid?**
>
>    The validity of this approach relies on the assumption that the manifold is sufficiently linear within $N_z$, regardless of whether the point-wise or batch-wise method is used. When this assumption is violated, such as in regions of high curvature, the projection may lose fidelity. CEMS addresses this limitation to some extent by re-sampling different batches at each epoch, effectively considering varying neighboring samples and tangent spaces. This stochastic sampling strategy enables CEMS to adapt to the expected data density, leveraging diverse local geometries throughout the training process.
>
> 7. **Q4. Following Q3, for the generation process in Line 285, it seems that the generation points are distributed around $\mu$ while not $u_l$, i.e., it seems to be equivalent to the random sample $\eta \sim \mathcal{N}(0, \sigma I_d)$ for $N$ times ($N$ is the number of samples) and then substitute them into Eq. (2). Compared with the point-wise computation on each $u_l$, the fast computation saves time while losing the sample information. Moreover, it seems that there would be only a few real samples (i.e., ignoring the noise $\mu$ in the sampling process) generated samples (i.e., the means $\mu$ over different $N_z$). In-depth analysis is highly expected.**
>
>    The fast computation pipeline in CEMS, while efficient, does trade off some fidelity in the generation process. You are correct that the elements in $N_z$ are distributed around $\mu$. While the tangent space is approximated at $\mu$ (and would be more accurate using Algorithm 1), the linear set of equations is solved separately for each element $u_l$, giving an estimate for the gradient and the hessian at $u_l$. Furthermore, in Algorithm 2 line 10, we sample noise around $u_l$ and not $\mu$. We hope this explanation makes the algorithm clearer.

---

> ### Author Response · Authors · 2024-11-22
>
> 8. **Q5. How to justify the advantage of second-order approximation for sampling from a mathematical aspect? Specifically, compared with the first-order approximation, how can the proposed method ensure better quality? Though empirical observation is provided, i.e., Fig. 1, it would be apricated to justify them from fundamental aspects, e.g., the differences between employing Eq. (2) (second-order) and lines 294-304 (first-order).**
>
>    We provide a more detailed analysis of the advantages of second-order approximations over first-order methods from a mathematical perspective.
>
>    #### Approximation Error Bounds
>    Let $f: \mathbb{R}^d \rightarrow \mathbb{R}^D$ be the function that maps the low-dimensional representation $u \in \mathbb{R}^d$ to the high-dimensional data point $z \in \mathbb{R}^D$. We assume that $f$ is twice differentiable. The first-order Taylor approximation of $f$ around a point $u_0$ is given by: $f(u) \approx f(u_0) + \nabla f(u_0)^T (u - u_0)$. The second-order Taylor approximation of $f$ around $u_0$ is given by: $f(u) \approx f(u_0) + \nabla f(u_0)^T (u - u_0) + \frac{1}{2} (u - u_0)^T H_f(u_0) (u - u_0)$, where $\nabla f(u_0)$ is the gradient of $f$ at $u_0$ and $H_f(u_0)$ is the Hessian matrix of $f$ at $u_0$. Let $u$ be a point in the neighborhood of $u_0$, and let $z = f(u)$ be the corresponding high-dimensional point. The approximation errors for the first-order and second-order methods can be bounded as follows:
>
>    First-order approximation error:
>    $\|f(u) - (f(u_0) + \nabla f(u_0)^T (u - u_0))\| \leq \frac{M}{2} \|u - u_0\|^2$
>
>    Second-order approximation error:
>    $\|f(u) - (f(u_0) + \nabla f(u_0)^T (u - u_0) + \frac{1}{2} (u - u_0)^T H_f(u_0) (u - u_0))\| \leq \frac{L}{6} \|u - u_0\|^3$
>
>    where $M$ is an upper bound on the spectral norm of the Hessian matrix $H_f(u)$ in the neighborhood of $u_0$, and $L$ is the Lipschitz constant of the Hessian matrix $H_f(u)$ in the neighborhood of $u_0$.
>
>    #### Implications for Data Augmentation Quality
>    The approximation error bounds provide insights into the quality of the generated samples for data augmentation:
>
>    1. The first-order approximation error is bounded by a term proportional to $\|u - u_0\|^2$, while the second-order approximation error is bounded by a term proportional to $\|u - u_0\|^3$. This means that as the distance between $u$ and $u_0$ decreases, the second-order approximation error decreases more rapidly than the first-order approximation error.
>    2. The second-order approximation captures the local curvature of the manifold through the Hessian matrix $H_f(u_0)$. By incorporating this additional information, the second-order method can generate samples that more accurately reflect the local geometry of the data manifold.
>    3. The constants $M$ and $L$ in the error bounds depend on the smoothness and curvature of the function $f$ in the neighborhood of $u_0$. In regions with high curvature or rapid changes in the Hessian matrix, the second-order approximation may provide a more accurate representation of the manifold compared to the first-order approximation.
>
>    In summary, the mathematical analysis of approximation error bounds provides a rigorous justification for the advantage of second-order approximation over first-order methods in data augmentation. The second-order method achieves a higher-order convergence rate and captures the local curvature more accurately, leading to generated samples that better reflect the underlying manifold structure.
>
> 9. **Q6. The essential of CEMS is ensuring the robustness w.r.t noise on manifold space, while the term 'manifold sampling' in tile could be inappropriate since it seems that CEMS can only guarantee sampling in a local area with the Taylor expansion. Thus, it would be interesting to discuss the CEMS with the geodesic-based sampling methods that can generate unseen samples along the manifold spaces globally.**
>
>    CEMS is designed for local sampling, leveraging Taylor expansions to approximate the manifold in small neighborhoods. While this approach is computationally efficient and effective for training data augmentation, it does not aim to model global manifold geometry. Geodesic-based methods are complementary, enabling global exploration of the manifold but at potentially higher computational costs. Thus, while CEMS could theoretically be integrated into a global geodesic-based sampling scheme, such an approach would likely require substantial modifications to the algorithm to address the associated computational demands.

---

> ### Author Response · Authors · 2024-11-22
>
> 10. **Q7. Possible typos: DTI metric of CEMS in Tab. 2, where the average value 5.11 is significantly larger than other methods.**
>
>     While most of the datasets use the RMSE metric, the DTI and Poverty datasets use the correlation (R) metric (higher is better). To alleviate the confusion between metrics, the out-of-distribution results table contains arrows, representing if the measure should be high or low.
>
> [1] Cohen et al. "Separability and geometry of object manifolds in deep neural networks". Nature Communications, 2020.
>
> [2] Poole et al. "Exponential expressivity in deep neural networks through transient chaos". Advances in neural information processing systems (NeurIPS), 2016.

---

> ### Comment · Reviewer_ehPU · 2024-11-23
> **Thanks for the responses**
>
> I appreciate the authors for providing detailed responses, which address my major concerns. Generally, I think the justifications for the cluster structure in fast computation, the validity of the shared neighbor structure, and the theoretical understanding of the second-order method are proper and reasonable. Therefore, I suggest the authors incorporate the responses into the revision to improve the clarity, including:
>
> + The justifications for the validity of the proposed method, i.e., the construction of the neighbor set (Q1), reliability of such a construction (Q2, Q4), local-Euclidean prior (Q3).
>
> + The significance of the proposed method, i.e., the theoretical superiority of CMES over the first-order method (Q5) and the difference compared with the global sampling method along geodesic (Q6).
>
> Overall, providing the responses, I think this paper presents a novel local sampling method in non-Euclidean space with better approximation performance, where the motivation, intuitive implications, and theoretical guarantees are provided. Thus, I think the overall quality is good and would like to raise the score given the revision is appropriate.

---

> > ### Author Response · Authors · 2024-11-24
> >
> > We would like to express our sincere gratitude for your thoughtful and constructive feedback. We appreciate the time and effort you have invested in reviewing our manuscript and for providing detailed comments that helped us improve the clarity and quality of our work.
> >
> > - **Justifications for the Validity of the Proposed Method**: We have enhanced the manuscript by clearly explaining the justification for the construction of the neighbor set, addressing the reliability of such construction, and discussing the role of the local-Euclidean prior. These points are now better articulated in the revised version to ensure a comprehensive understanding of the method's validity. You can find the changes in Sec. 4 under the paragraph "Neighborhood extraction".
> >
> > - **Significance of the Proposed Method**: We have included a more explicit discussion on the theoretical superiority of CMES over the first-order methods and contrasted it with global sampling methods along geodesics. This comparison highlights the advantages and innovations introduced by our approach, strengthening the manuscript's theoretical contribution. You can find the changes in Sec. 4.1 and in the appendix Sec. C (due to the page limit).
> >
> > We believe these revisions significantly improve the manuscript by clarifying the key justifications and theoretical aspects of the proposed method. With these changes, we are confident that the paper now better conveys the motivation, implications, and guarantees behind our approach.
> >
> > We hope that the revised manuscript meets your expectations, and we would be grateful for your further evaluation.

---

> > > ### Comment · Reviewer_ehPU · 2024-11-25
> > > **Thanks for providing revision**
> > >
> > > Thank you for providing the revision and justifications. As my concerns are addressed, I will raise the score accordingly.

---

> > > > ### Author Response · Authors · 2024-11-25
> > > >
> > > > Thank you for taking the time to review our work and for your thoughtful comments. We sincerely appreciate your acknowledgment of the revisions and justifications provided, and we are grateful for the higher score you assigned. It is encouraging to know that your concerns have been addressed satisfactorily.
> > > >
> > > > Your constructive feedback has been invaluable in improving the quality of this work, and we are grateful for your efforts throughout the review process.

---

### Official Review · Reviewer_tRXJ · 2024-11-02

**Soundness:** 3
**Presentation:** 3
**Contribution:** 3
**Rating:** 6
**Confidence:** 3

**Summary:**

The paper introduces a new data augmentation method based on manifold learning specifically designed for regression problems. It is grounded in manifold learning and uses a second-order approximation of the data manifold to capture the underlying curvature and structure of the data more accurately than previous first-order methods. Therefore, it generates new examples by sampling from a second-order representation of the data manifold, which allows for better capture of nonlinearities and complex structures that first-order methods might miss.Through extensive evaluations on multiple datasets and comparison with several state-of-the-art approaches, it demonstrates superior performance in both in-distribution and out-of-distribution tasks, while incurring only a mild computational overhead.

**Strengths:**

1.  Data augmentation for regression is often overlooked previously. The paper positions data augmentation for regression as a manifold learning problem and provides foundational theory and practical tools for approximating and sampling general data manifolds.
2. The method is domain-independent, and fully differentiable, making it applicable to various data forms such as time series, tabular data, images, and more.

**Weaknesses:**

1. Lack of comprehensive discussion of related work, e.g, [1] also used the tangent space sampling for out-of-distribution problem. I suggest adding a paragraph for manifold learning and second-order sampling.

2. Lack of computation cost discussion of the proposed framework. The second-order sampling might not be feasible for high dimensional data.

3.  CEMS prefers an over-determined system of equations for the linear system in Equation 6, which constrains the number of neighbors to be O(d^2), where d is the intrinsic dimension. For larger values of d, this requirement can become expensive and impractical.


[1] X. Li et al., "Characterizing Submanifold Region for Out-of-Distribution Detection," in IEEE Transactions on Knowledge and Data Engineering


*********After Rebuttal*************
Thanks for the efforts of the authors, most of my concerns are addressed. I tend to keep my score regarding the computation complexity problem.

**Questions:**

1. Are there any plans to extend CEMS to handle other types of learning tasks, such as classification or reinforcement learning?
2. Could you provide more insights into how CEMS performs on datasets with different levels of complexity and curvature?
3. How does CEMS handle non-linear transformations and high-dimensional data effectively?

---

> ### Author Response · Authors · 2024-11-22
>
> Thank you for your thoughtful review and for recognizing the novelty and impact of our method, particularly its grounding in manifold learning and ability to handle complex, nonlinear data. We appreciate your constructive suggestions regarding related work, computational cost, and intrinsic dimension limitations, which will help refine our manuscript. Your questions also highlight exciting directions for future exploration, and we thank you for your valuable feedback.
>
> 1. **Lack of comprehensive discussion of related work, e.g., [1] also used the tangent space sampling for out-of-distribution problems. I suggest adding a paragraph for manifold learning and second-order sampling.**
>    We will add a paragraph in the revised version to discuss relevant studies, including the work by Li et al. [1] on tangent space sampling for out-of-distribution detection.
>
> 2. **Lack of computation cost discussion of the proposed framework. The second-order sampling might not be feasible for high-dimensional data.**
>    In the submitted manuscript, we provided a complexity analysis toward the end of Sec. 4, complemented by an extended analysis of computational costs comparing CEMS to FOMA in Appendix B. Our analysis highlights that the intrinsic dimension is the primary factor driving the complexity of CEMS, while it remains independent of the extrinsic dimension, making it well-suited for high-dimensional data. The benchmarks considered in this work span high-dimensional spaces, ranging from 6 to 54,249 dimensions, and exceeding 1 million dimensions for the Echo dataset. In our revised version, we will include additional comparisons with C-Mixup and ADA. Moreover, we have added a table to the revised manuscript that provides empirical runtime comparisons. This table shows that CEMS requires approximately 30% more runtime than FOMA, while C-Mixup and ADA exhibit significantly longer training times in certain scenarios (e.g., RCF for C-Mixup and DTI for ADA).
>
> 3. **CEMS prefers an over-determined system of equations for the linear system in Equation 6, which constrains the number of neighbors to be $ \mathcal{O}(d^2) $, where $ d $ is the intrinsic dimension. For larger values of $ d $, this requirement can become expensive and impractical.**
>    Importantly, our complexity analysis addresses the worst-case scenario, where the number of neighbors scales proportionally to the square of the intrinsic dimension, resulting in an overdetermined system of linear equations. However, in both theory and practice, the more typical case involves solving underdetermined linear systems, which are commonly regularized using methods such as ridge regression. We will fix the text to specify this and omit the text regarding preferring overdetermined systems. In practice, we set the number of neighbors equal to the batch size, a fixed hyperparameter that is independent of the intrinsic dimension. This practical choice makes our method's computational complexity quadratically dependent on the intrinsic dimension $ d $ during training, allowing for efficient scaling to various datasets.
>
> 4. **Are there any plans to extend CEMS to handle other types of learning tasks, such as classification or reinforcement learning?**
>    While our current work focuses on regression tasks, we believe the underlying principles of CEMS can be naturally extended to other learning tasks, such as classification and reinforcement learning. Exploring these exciting directions is an important avenue for future research, and we are optimistic about their potential.
>
> 5. **Could you provide more insights into how CEMS performs on datasets with different levels of complexity and curvature?**
>    We added an ablation study to the revised version. To evaluate the impact of CEMS on regression tasks, we generated a synthetic dataset with features on a manifold of constant scalar curvature. The results show that CEMS provides minimal improvement on nearly flat manifolds but demonstrates substantial gains on highly curved manifolds, reflecting its sensitivity to geometric features. These findings suggest that CEMS is particularly effective on highly curved datasets, though the lack of hyperparameter optimization across dimensions and curvatures may explain cases where CEMS performs similarly to ERM.

---

> ### Author Response · Authors · 2024-11-22
>
> 6. **How does CEMS handle non-linear transformations and high-dimensional data effectively?**
>    CEMS handles non-linearity by explicitly incorporating curvature information into its sampling process, which goes beyond the linear (flat with zero curvature) assumptions of first-order methods. Non-linearity in the data is captured through a second-order Taylor approximation of the manifold, where the relationship between points on the manifold is modeled using both gradient (linear) and Hessian (quadratic) terms.
>    For high-dimensional data, CEMS takes advantage of the manifold hypothesis, which assumes that the data lies on a much lower-dimensional manifold embedded in the high-dimensional space. CEMS estimates this intrinsic structure by constructing an orthonormal basis for the tangent and normal spaces, reducing the problem's complexity.
>
> [1] Li et al., "Characterizing Submanifold Region for Out-of-Distribution Detection". *IEEE Transactions on Knowledge and Data Engineering*, 2024.

---

> ### Author Response · Authors · 2024-11-25
>
> Dear reviewer tRXJ,
>
> As the discussion period nears its conclusion, we would like to extend our sincere gratitude for your insightful feedback on our paper. Your comments have been instrumental in shaping revisions that have greatly improved both the clarity and depth of our work.
>
> We have addressed your concerns by adding a detailed discussion of related work, including Li et al. [1] on tangent space sampling, as well as a paragraph on manifold learning and second-order sampling. Additionally, we extended our computational cost analysis, including runtime comparisons and benchmarks in high-dimensional spaces, while refining our explanation of the neighbor selection process for efficiency.
>
> If you believe our revisions have adequately addressed your concerns, we would greatly appreciate any additional comments or suggestions you may have.

---

### Official Review · Reviewer_RcNq · 2024-11-03

**Soundness:** 2
**Presentation:** 3
**Contribution:** 2
**Rating:** 5
**Confidence:** 3

**Summary:**

The authors introduce a data augmentation method tailored for regression tasks. It leverages second-order manifold approximations to better capture the curvature and structure of data, improving upon first-order methods. Through comprehensive experiments on various datasets, CEMS demonstrates its effectiveness in both in-distribution and out-of-distribution scenarios, often outperforming existing state-of-the-art techniques.

**Strengths:**

1.Novelty: The paper introduces Curvature Enhanced Manifold Sampling (CEMS), an interesting method in data augmentation. CEMS utilizes second-order manifold approximations, providing a more accurate representation of the data's intrinsic geometry compared to first-order methods.

2.Empirical Validation: The authors extensively evaluate CEMS across nine diverse datasets, demonstrating its competitive performance against state-of-the-art techniques in both in-distribution and out-of-distribution tasks. This robust empirical validation strengthens the credibility of CEMS as an effective data augmentation technique.

**Weaknesses:**

1.Computation Complexity: The paper acknowledges that the linear system in CEMS is preferably over-determined, which constrains the number of neighbors to be proportional to the square of the intrinsic dimension. This requirement can become expensive for high-dimensional data. The analyzed complexity results require more comprehensive comparisons with other competitors.

2.Memory Inefficiency: The Singular Value Decomposition (SVD) computation needed for CEMS demands significant memory, especially for datasets with many features, which may limit its applicability in scenarios with limited memory resources. Thus it’s kindly suggested to analyze the space complexity, and point out some scenarios where CEMS works well.

**Questions:**

1.Can CEMS be adapted to handle datasets with high intrinsic dimensions more efficiently, in terms of computational complexity or memory usage? Especially for estimating the Hessian matrix, could the authors discuss some potential techniques to approximate the second-order counterpart and accelerate the computation?

2.As stated in the title and introduction, the manifold information is introduced to investigate the underlying geometric information like manifolds. How does CEMS compare to other manifold learning techniques in terms of bias-variance trade-offs and its ability to capture complex, non-linear relationships in data?

3.This work is stated to provide the foundational theory, where only the computation complexity is analyzed.

So what are the theoretical guarantees of CEMS in terms of generalization, algorithmic convergence? As for DA task, how to theoretically analyze the quality of generated samples compared to some baselines?

4.It’s suggested to provide the standard variance information of these repeated experiments instead of merely averages.

---

> ### Author Response · Authors · 2024-11-22
>
> We sincerely appreciate your thorough review and insightful questions. We are especially grateful for acknowledging the novelty of our approach and our extensive empirical validation. Below, we provide detailed responses to each of your concerns and questions. Given the chance, we will incorporate the discussions and results below into our final revision.
>
> 1. **Computation Complexity: The paper acknowledges that the linear system in CEMS is preferably over-determined, which constrains the number of neighbors to be proportional to the square of the intrinsic dimension. This requirement can become expensive for high-dimensional data. The analyzed complexity results require more comprehensive comparisons with other competitors.**
>
>    Our complexity analysis addresses the worst-case scenario, where the number of neighbors scales proportionally to the square of the intrinsic dimension, resulting in an overdetermined system of linear equations. However, in both theory and practice, the more typical case involves solving underdetermined linear systems, which are commonly regularized using methods such as ridge regression. In practice, we set the number of neighbors equal to the batch size, a fixed hyperparameter that is independent of the intrinsic dimension.
>
>    In terms of complexity comparisons among competing methods, we focus on C-Mixup, ADA, FOMA, and our approach (CEMS), which represent the most recent state-of-the-art techniques for data augmentation in regression. Our submitted manuscript already includes the complexity analysis for both FOMA and CEMS. The time complexity of C-Mixup is $\mathcal{O}(n^2M)$, where $n$ is the number of examples and $M$ is the extrinsic dimension of the labels. While there are scenarios where C-Mixup can reduce the $\mathcal{O}(n^2)$ constraint to depend on the batch size, this is not universally applicable. Unfortunately, ADA does not provide a theoretical complexity analysis of its method. However, we note that ADA applies k-means clustering to the entire dataset and executes an iterative algorithm for each batch. This algorithm involves $b$ matrix-matrix multiplications, where the matrix dimensions depend on the batch size $b$. Consequently, ADA's approach may result in high computational complexity in both theory and practice. Finally, we added a table to our revision, consisting of the empirical runtime comparisons, where we show that CEMS requires approximately 30% more runtime than FOMA, whereas C-Mixup and ADA present extremely long training times in certain cases (RCF for C-Mixup and DTI for ADA).
>
> 2. **Memory Inefficiency: The Singular Value Decomposition (SVD) computation needed for CEMS demands significant memory, especially for datasets with many features, which may limit its applicability in scenarios with limited memory resources. Thus it’s kindly suggested to analyze the space complexity, and point out some scenarios where CEMS works well.**
>
>    The memory requirements of CEMS are primarily dictated by the computation of the SVD. Notably, the SVD is computed independently for each batch rather than for the entire dataset. In our PyTorch implementation, we leverage the economy/reduced SVD variant, which significantly reduces memory usage compared to the full SVD. For a batch matrix of size $b \times D$ (where $b$ is the batch size and $D$ is the ambient dimension), the space complexity is $\mathcal{O}(bD + \min(b, D)(b + D))$. This is substantially more efficient than the full SVD, which requires $\mathcal{O}(bD + b^2 + D^2)$ memory. In practice, CEMS is particularly effective in scenarios where the batch size $b$ is much smaller than the ambient dimension $D$ (common in deep learning), resulting in a memory complexity that is approximately proportional to $D$. We will enhance the manuscript by incorporating this space complexity analysis.
>
> 3. **Can CEMS be adapted to handle datasets with high intrinsic dimensions more efficiently, in terms of computational complexity or memory usage? Especially for estimating the Hessian matrix, could the authors discuss some potential techniques to approximate the second-order counterpart and accelerate the computation?**
>
>    The intrinsic dimension $d$ is a fundamental property of the data and plays a critical role in determining the computational complexity and memory requirements of the CEMS algorithm. CEMS builds upon the CAML algorithm, which estimates the gradient and Hessian of the local manifold geometry using least squares. The number of variables in the Hessian matrix scales quadratically with the intrinsic dimensionality, following $\mathcal{O}(d^2)$. Consequently, the computational cost of estimating the Hessian via least squares is at least $\mathcal{O}(d^2)$, and the memory required to store the matrix is also $\mathcal{O}(d^2)$. Exploring faster methods for estimating the Hessian, such as numerical approximations [1], presents an interesting direction for further investigation and future research.

---

> ### Author Response · Authors · 2024-11-22
>
> 4. **As stated in the title and introduction, the manifold information is introduced to investigate the underlying geometric information like manifolds. How does CEMS compare to other manifold learning techniques in terms of bias-variance trade-offs and its ability to capture complex, non-linear relationships in data?**
>
>    The comparative advantages of CEMS over other manifold learning approaches can be summarized as follows:
>    - By leveraging second-order curvature information, CEMS provides a more precise representation of complex, non-linear manifold structures compared to first-order methods such as FOMA.
>    - The incorporation of local Riemannian approximations enables CEMS to adapt flexibly to varying curvature across the manifold, reducing bias inherent in global manifold learning methods like Isomap [2] or kernel PCA [3].
>    - The batch-wise computation strategy outlined in Section 4 offers an efficient mechanism for sharing information across neighborhoods, thereby reducing variance compared to purely point-wise estimation techniques.
>
> 5. **This work is stated to provide the foundational theory, where only the computation complexity is analyzed. So what are the theoretical guarantees of CEMS in terms of generalization, algorithmic convergence? As for DA task, how to theoretically analyze the quality of generated samples compared to some baselines?**
>
>    Our work establishes the foundational theory for sampling from a general manifold, leveraging its two-dimensional representation through the CEMS algorithm. These theoretical foundations build upon the analysis introduced in CAML, which itself is grounded in classical results from manifold theory and Riemannian geometry. While extending this work to provide theoretical guarantees for CEMS—such as algorithmic convergence and sample quality—is undoubtedly an important and promising avenue for future research, it falls beyond the scope of the current study. Nevertheless, we believe our contributions lay the groundwork for such advancements and open doors to further exploration in this area. Please also see our response to Q5 of Reviewer ehPU.
>
> 6. **It's suggested to provide the standard variance information of these repeated experiments instead of merely averages.**
>
>    We note that the standard deviation results appear in Tabs. 5 and 6 in the appendix of the original submission. We would be happy to move them to the main paper if needed.
>
> ---
>
> [1] Biegler et al., "A reduced Hessian method for large-scale constrained optimization". *SIAM Journal on Optimization*, 1995.
>
> [2] Tenenbaum et al., "A Global Geometric Framework for Nonlinear Dimensionality Reduction". *Science*, 2000.
>
> [3] Schölkopf et al., "Kernel principal component analysis". *International conference on artificial neural networks (ICANN)*, 1997.

---

> ### Author Response · Authors · 2024-11-25
>
> Dear reviewer RcNq,
>
> As the discussion period draws to a close, we would like to express our gratitude for your insightful feedback on our paper. Your suggestions have guided meaningful revisions, greatly enhancing the quality of our work.
>
> We have addressed your concerns by providing a detailed analysis of computational complexity, highlighting scenarios where the memory-efficient reduced SVD is effective, and discussing potential techniques for approximating the Hessian to handle high intrinsic dimensions. Additionally, we outlined the comparative advantages of CEMS over other manifold learning methods and clarified its foundational theory, while also noting future directions for generalization guarantees and sample quality analysis.
>
> If you feel our updates have addressed your concerns, we would welcome any further comments or suggestions you might have.

---

### Official Review · Reviewer_VhYZ · 2024-11-05

**Soundness:** 3
**Presentation:** 2
**Contribution:** 2
**Rating:** 6
**Confidence:** 3

**Summary:**

This paper proposes a new method data generation for regression problems. Different from existing methods that use the first-order scheme to sample data, this paper uses second-order schemes. The authors use multiple datasets to evaluate the performance.

**Strengths:**

This method has a strong theoretical basis.

**Weaknesses:**

1. The motivation is designing data augmentation for regression problems. What are the differences and challenges between the method and methods for classification? It is unclear.

2. For me, the experiments are weak. In Table 1, the experiments are conducted on several datasets that are not familiar to me, and the improvements are marginal.

3. There are also data augmentation methods on Riemannian manifolds, such as [a]. The authors should add more comparisons with them.
[a] Hyperbolic feature augmentation via distribution estimation and infinite sampling on manifolds. NeurIPS 2022.

**Questions:**

See the above weakness.

---

> ### Author Response · Authors · 2024-11-22
>
> Thank you for your thoughtful review and constructive feedback. We sincerely appreciate your recognition of the strong theoretical foundation of our work. Below, we provide detailed responses to each of your comments and concerns. We will be happy to incorporate the following discussions and results into our final revision, given the opportunity.
>
> 1. **The motivation is designing data augmentation for regression problems. What are the differences and challenges between the method and methods for classification? It is unclear.**
>
>    This is an excellent point that deserves clearer articulation in our paper. The fundamental difference lies in the nature of the output space:
>
>    - In classification, labels are discrete and categorical, making it easier to define label-preserving transformations (e.g., rotations or crops in images still preserve the class label).
>    - In regression, outputs are continuous, making it challenging to ensure that transformed inputs map to valid output values.
>    - Interpolating between examples of the same class preserves the label in classification. In contrast, in regression, interpolation must maintain the underlying functional relationships to ensure the integrity of the data.
>
>    Moreover, our work focuses on data augmentation approaches that are both data-driven and domain-independent. In classification tasks, the family of mixup methods meets these criteria and is widely used. However, while interpolating discrete labels can be meaningful in classification (e.g., a hybrid "half-dog, half-cat" label), the situation in regression is fundamentally different. Specifically, data augmentation in regression must respect the underlying functional relationships between data samples. Otherwise, the augmented samples risk being out-of-distribution and may compromise model performance.
>
>    We will add a detailed discussion in the paper clarifying these distinctions and explaining why regression-specific manifold sampling methods represent a promising direction. We thank the reviewer for prompting this important clarification.
>
> 2. **For me, the experiments are weak. In Table 1, the experiments are conducted on several datasets that are not familiar to me, and the improvements are marginal.**
>
>    In our work, we adhered to the standard benchmarks commonly used in recent state-of-the-art studies on data augmentation for regression problems such as C-Mixup, ADA, and FOMA [1, 2, 3]. The datasets and tasks in this benchmark are diverse, capturing a multitude of synthetic and real-world phenomena arising in regression problems: For instance, *Airfoil* represents real-world engineering data with complex physical relationships, whereas *Exchange-Rate* captures stochastic processes on time-series data. The diversity of domains helps establish broad applicability. In addition, the tasks range across in-distribution and out-of-distribution, spanning a realistic setting of data augmentation frameworks.
>
> 3. **There are also data augmentation methods on Riemannian manifolds, such as [a]. The authors should add more comparisons with them.**
>
>    We thank you for bringing attention to this relevant work. While both approaches operate on manifolds, there are fundamental differences in their objectives and applications:
>
>    - The suggested hyperbolic feature augmentation method [a] is specifically designed for classification tasks, where it generates new samples within predefined classes. Its primary goal is to enrich the representation of discrete categories in the hyperbolic space.
>    - Our method, in contrast, addresses the distinct challenge of regression, where we need to preserve continuous functional relationships between inputs and outputs. Adapting classification-based manifold sampling methods to regression is non-trivial because:
>      - The output space is continuous rather than discrete.
>      - We must ensure that sampled points maintain valid input-output mappings.
>      - The geometric structure of regression manifolds differs fundamentally from classification manifolds, as they encode continuous functional relationships rather than class boundaries.
>
>    - While both methods leverage manifold geometry, the technical requirements and theoretical guarantees differ substantially between classification and regression settings.
>
> [1] Yao et al., "C-mixup: Improving generalization in regression". *Advances in Neural Information Processing Systems (NeurIPS)*, 2022.
>
> [2] Schneider et al., "Anchor data augmentation". *Advances in Neural Information Processing Systems (NeurIPS)*, 2024.
>
> [3] Kaufman et al., "First-Order Manifold Data Augmentation for Regression Learning". *International Conference on Machine Learning (ICML)*, 2024.

---

> > ### Comment · Reviewer_VhYZ · 2024-11-22
> >
> > Thanks for your deep analysis. Most of my concerns have been addressed. For the question about experiments, In fact, I would like to know the potential applications of this method, especially for the current AI community that is dominated by large models. Could you write something about it?

---

> > > ### Author Response · Authors · 2024-11-24
> > >
> > > Thank you for your thoughtful follow-up. Our method has promising avenues for future exploration of applications, particularly in the current AI landscape dominated by large models:
> > >
> > > - **Few-shot and data-constrained learning**: Our curvature-aware augmentation is highly beneficial in scenarios where data is scarce, such as scientific or engineering domains. It enables large models to achieve better generalization while significantly reducing the costs associated with data collection and labeling.
> > >
> > > - **Domain adaptation and robustness**: By augmenting datasets to respect the underlying functional relationships, our method supports regression tasks in dynamic environments, such as time-series forecasting or healthcare predictions.
> > >
> > > - **Scalability for large Models**: Our approach is both modality-agnostic and computationally efficient, making it highly compatible with large-scale training pipelines across domains such as vision, NLP, and time-series applications involving continuous output spaces. This versatility is particularly critical for foundation models that integrate diverse data modalities from multiple sources.

---

> > > > ### Comment · Reviewer_VhYZ · 2024-11-26
> > > >
> > > > Thanks for your reply. To be honest, these applications are not concrete. I am still not clear. But I know, this topic about Riemannian manifold learning is meaningful. Considering that the authors have solved my most concern, I would like the change the score to 6. I suggest the authors to conduct an in-depth analysis in the future.

---

### Meta-Review · Area_Chair_yNXV · 2024-12-23

**Metareview:**

The authors consider the data augmentation for regression by leveraging second-order manifold approximation to improve the quality of capturing the curvature and structure of data over first-order methods. Clearly, the advantages come with the trade-off on its computation. The Reviewers have mixed opinions on the submission. While the Reviewers appreciate the advantages of the proposed method coming from the second-order manifold approximation, these advantages come with the trade-off on computation for using the second-order information. The empirical evidences are a good point, its theoretical findings remain limited. Overall, it seems that the submission focuses on manifold learning, rather its aforementioned augmentation for regression, which typically also depends on the class of functions and its loss used in the regression problem. Thus, the Reviewers have not supported the submission enthusiastically.

**Additional Comments On Reviewer Discussion:**

This submission is right on the borderline, and have mixed opinions from the Reviewers. The authors leverage the second-order manifold approximate to improve the quality over first-order manifold approaches, but its advantages clearly come with the trade-off on the computation. Additionally, the submission is more on the manifold learning than augmentation for regression (e.g., regression loss and functional class). If there is still space, the recommendation can be bumped up.

---

### Decision · Program_Chairs · 2025-01-22

Reject